# Dams threaten salmonids by triggering temperature-dependent proliferative kidney disease
Magnus Lauringson [1], Joacim Näslund [2], Lilian Pukk [1], Siim Kahar[1], Riho Gross[1] & Anti Vasemägi[1,2] ✉

Dams provide different services for human society, but they also significantly disrupt ecosystems by altering natural flow and temperature regimes. Here, we describe a novel, unappreciated threat posed by reservoirs to one of the world's most popular game fish, brown trout (*Salmo trutta*). We show that small river impoundments elevate downstream water temperature during summer, which increases the prevalence and abundance of *Tetracapsuloides bryosalmonae* parasite triggering proliferative kidney disease (PKD), an emerging disorder in salmonids across North America and Europe. Our study highlights the role of reservoirs in creating parasite and disease hotspots, while providing limited evidence that dams act as barriers to parasite spread. This makes downstream areas from reservoirs valuable sentinel sites for monitoring climate impacts on riverine ecosystems. Ultimately, the assessment of dams requires a more holistic approach, where the disease risks are included in the decision-making process balancing human needs with the health of aquatic ecosystems.

Building dams to manipulate running water is as old as human civilization. Recent estimates suggest that millions of dams have been constructed worldwide[1] and their continued construction progresses at a high pace to provide renewable energy, water security and flood management, particularly in emerging economies[2,3]. However, these man-made barriers impede free flow of water, and their associated reservoirs have significant effects on water physiochemistry, biodiversity, and ecological connectivity, with far-reaching negative consequences for environmental sustainability[4,5]. Increased attention to environmental recovery has rendered removal of barriers, in particular obsolete ones, an important part of current river restoration activities, often with special emphasis on ecological connectivity and movement of aquatic organisms within river networks[6,7].

In addition to habitat fragmentation, climate change is increasingly threatening river ecosystems by altering hydrological patterns and thermal regimes.[8] Dams exacerbate these impacts by disrupting the natural flow of water, nutrients and sediment, which can lead to eutrophication, algal blooms, oxygen depletion and release of greenhouse gases.[9] Their impact on downstream thermal conditions varies depending on river hydrology, reservoir characteristics, and, most critically, the depth from which water is released—either from the surface or the hypolimnion.[10] Small, surface-release dams, which constitute the majority of dams globally, can elevate summer water temperatures by several degrees, mirroring short-term projections of climate warming.[11,12]

While many negative environmental consequences of barriers in natural lotic ecosystems are widely recognised, we currently have very limited knowledge on how dams alter epidemiology of temperature-dependent diseases in aquatic wildlife as global temperatures continue to rise.[13,14] Given that the climate acts as an important driver of spatial and seasonal patterns of infections,[14] reservoirs that accumulate heat during the hot summer months[11,12] can drive disease outbreaks. Alternatively, dams may function as barriers for upstream pathogen spread, with potential positive implications for reducing the risk of system-wide disease outbreaks.[15] Despite their significant ecological implications, the role of pathogens and diseases remains largely overlooked by decision-makers and environmental managers when setting priorities for dam removal or restoration of floodplains.

In this study, we focus on an emerging temperature-dependent disease in salmonid fish (Proliferative Kidney Disease; PKD), caused by the malacosporean *Tetracapsuloides bryosalmonae* (*Tb*). This microparasite is widely distributed in North America and Europe, and PKD has been implicated in the decline of salmonid populations in many countries.[16] The parasite's lifecycle alternates between colonial bryozoans as its primary host and salmonids as the vertebrate vector. Parasite development in bryozoans is positively correlated with temperature and nutrient content of water.[17] Furthermore, severity and fatality of PKD in salmonids is strongly temperature-dependent.[18,19] Resulting disease symptoms, such as enlargement of the kidney (granulomatous nephritis with vascular necrosis) and

¹Chair of Aquaculture, Institute of Veterinary Medicine and Animal Sciences, Estonian University of Life Sciences, Tartu, Estonia. ²Department of Aquatic Resources, Institute of Freshwater Research, Swedish University of Agricultural Sciences, Drottningholm, Sweden. ✉e-mail: anti.vasemagi@slu.se

anemia, leads to high mortality when the water temperature exceeds 15–17 °C over an extended period of time.[18,20] Given that parasite lifecycle, host resilience, and disease outcome are all temperature-dependent, the impact of PKD on salmonid populations is expected to exacerbate with global warming.[21] However, there is currently a gap in understanding the potential role of dams and associated reservoirs in triggering PKD, and more broadly, in contributing to the rise of temperature-dependent diseases in aquatic wildlife.

We investigated the impact of man-made dams on parasite abundance and disease severity by examining juvenile brown trout (*Salmo trutta*) from up- and downstream of small reservoirs and artificial impoundments. To assess the role of dams as catalysts of PKD or alternatively, as migration barriers for parasite spread, we measured *Tb* infection prevalence and abundance (parasite load), together with main disease symptoms (renal hyperplasia and anemia) up- and dowstream of dams and reservoirs. We focused on brown trout recruitment areas located as close to the dams as possible, in order to minimise the influence of natural environmental factors, such as downstream warming or coldwater inputs from groundwater and tributaries, that could confound the assessment of dam and reservoir effects on PKD. Our findings provide compelling evidence that reservoirs function as catalysts for temperature-driven disease in brown trout, while providing limited evidence that dams act as barriers for *Tb* spread. Notably, lotic environments downstream of small reservoirs emerge as parasite and disease hotspots, rendering salmonids particularly vulnerable to rising temperatures.

## Results
### Water temperature and body size
Compared to upstream sampling sites, the water temperature was consistently higher downstream of the reservoirs in all studied locations ($N = 28$ locations, 14 pairs/rivers, Fig. 1A, D, and Supplementary Table 1). On average, water temperature was 2.64 °C higher (95% CI: 1.69 to 3.58 °C, range: 0.24 to 5.73 °C) downstream of the dams, as compared to the upstream location (Wilcoxon signed-rank test df = 13, $U = 2$, $P < 0.001$, $N = 28$ locations; Fig. 1D). Similarly, downstream locations exhibited water temperature over 15 °C (clinical PKD threshold) for longer time period compared to upstream locations (average no. of days >15 °C, up- vs. downstream: 33.21 vs 51.3, Wilcoxon signed-rank test df = 13, $U = 0$, $P < 0.001$, $N = 28$ locations; Fig. 1E, and Supplementary Table 1). There was no significant difference in diurnal water temperature fluctuation between upstream and downstream locations (up- vs. downstream: 2.05 °C vs. 1.96 °C, Wilcoxon signed-rank test df = 13, $U = 42.5$, $P = 0.552$, $N = 28$ locations; Supplementary Fig. 1, and Supplementary Table 1). The extent of downstream warming was positively associated with reservoir size, though this did not reach statistical significance ($\beta$: 1.3576, 95% CI: 0.514 to 2.592, $P = 0.074$, LM; Fig. 1F). Body size did not differ significantly between locations upstream and downstream of dams ($N = 13$ dams, one dam not included due to lack of young-of-the-year (YOY) trout in downstream location, see under *Sampling of fish*; linear mixed model (LMM), factor: location; $\chi^2 = 0.03$, $P = 0.870$; Supplementary Table 2, $N = 442$ individuals). Body mass at a given length (body condition), however, was on average slightly lower (ca. 2.2%) in the upstream locations ($\beta_{upstream}$: −0.022 ± 0. 01 SE, $t = −2.83$, $P = 0.005$; Supplementary Table 3, and Supplementary Fig. 2).

### Infection prevalence and parasite load
A binomial generalized linear mixed model (GLMM) indicated that infection prevalence was significantly lower at upstream locations, as compared to downstream locations, when analysing the nine *Tb*-present rivers [factor: location; $\chi^2 = 42.1$, $P < 0.001$, $N = 18$ locations (9 pairs/rivers); Supplementary Table 4, infection prevalence model for all rivers presented in Supplementary Table 5]. However, the random effect variance was large (SD = 5.72) relative to the fixed effect ($\beta_{upstream} = -4.97$), indicating high variability among rivers (due to several extreme values, i.e. 100% prevalence). This variability inflates expected values (Fig. 2A), leading the model to overestimate mean prevalence. To confirm the model-indicated

result, we also compared mean prevalence between locations using a pairwise *t*-test, which also showed significantly lower upstream prevalence ($t = 2.6$, df = 8, $P = 0.029$, $N = 18$ locations).

Parasite load, quantified as the number of *Tb* 18S rRNA gene copies per quantitative PCR (qPCR) reaction (*Tb* copies/reaction), was generally higher among fish in downstream locations of the dams (Fig. 2B), with effects detected using both the full data set (LMM: $\beta_{upstream}$: −47.2 ± 5.1 SE, $t = −9.30$, $P < 0.001$, $N = 444$ individuals, Supplementary Table 6) and the reduced set excluding apparently *Tb*-absent rivers ($\beta_{upstream}$: −65.4 ± 6,8 SE, $t = −9.60$, $P < 0.001$, $N = 315$ individuals, Supplementary Table 7). The maximum parasite loads for each river are presented in Supplementary Fig. 3 and the average values (±standard error) for each river are presented in Supplementary Fig. 4. Using a pairwise *t*-test to compare parasite load of fish upstream and downstream locations in *Tb* present rivers, the differences are still significant ($t = 2.1$, df = 8, $P = 0.034$, $N = 18$ locations).

### Renal hyperplasia and anemia
Renal hyperplasia, indicated by a high K/B-ratio, here analysed as K at a given B, was more severe among fish in the downstream locations ($\beta_{downstream}$: 0.50 ± 0.04 SE, $t = 11.8$, $P < 0.001$, $N = 443$ individuals, Supplementary Table 8). From the follow-up model (Supplementary Table 9), the latter result was verified in the *Tb* present rivers [estimated difference at average body length (upstream-downstream): −0.64 mm, $t = −13.1$, $P < 0.001$], but not in *Tb* absent rivers (estimated difference: −0.14 mm, $t = −1.70$, $P = 0.323$). Fish in downstream locations in *Tb* absent rivers had lower levels of K/B-ratio than fish in downstream locations of *Tb* present rivers (estimated difference: −0.78 mm, $t = −3.22$, $P = 0.032$) and the same was the case in the upstream locations (estimated difference: −0.92 mm, $t = −3.82$, $P = 0.012$) (Fig. 2C, D, and Supplementary Fig. 4).

The K/B-ratio started to increase already at relatively low parasite loads (>1000 *Tb* copies/reaction) and the average K/B-ratio plateaued at parasite loads above ~8000 *Tb* copies/reaction (Fig. 3A). After plateauing, there was a substantial variation in K/B-ratio, but no values were found to be lower than the average for fish in *Tb* absent rivers.

Low hematocrit values (<0.25) were found among individuals with the highest parasite load (>40,000 *Tb* copies/reaction), with most heavily affected fish displaying severe anemia (hematocrit: <0.15, Fig. 3B). Hematocrit showed a clear negative relationship with K/B-ratio in *Tb* present rivers, which clearly deviates from the uninfected individuals (Fig. 3C). Only a subset of individuals ($N = 27$ individuals) with enlarged kidneys (K/B-ratio: >0.2) showed anemic response (hematocrit: <0.25).

## Discussion
The majority of rivers in Europe and North America are fragmented by numerous dams and weirs, which disrupt flow, habitat connectivity, and natural water temperature regime.[1,7] Our study reveals that surface-release dams amplify temperature-dependent disease with warmer lotic environments downstream of small reservoirs serving as parasite and disease hotspots. This adds further pressure on cold-water salmonid populations, which are increasingly threatened by the effects of climate change across their global range.[22,23]

### Reservoirs warm up water
We observed increased water temperature downstream of all studied dams, reflecting a widespread effect. Notably, the average water temperatures were up to 5.73 °C warmer than upstream locations (mean 2.64 °C). These values are comparable to reported summertime warming caused by small surface-release dams elsewhere.[12] However, unlike earlier studies, we did not measure temperature directly below the dam or upstream of the reservoir. Instead, we focused on brown trout recruitment areas to obtain water temperature data relevant for the disease progression. Consequently, the combined warming effect of the reservoir and associated disease risk may be even more pronounced immediately downstream of the dam.

Previous studies have shown that typical PKD symptoms, such as renal hyperplasia and anemia, emerge when the water temperature exceeds 15 °C

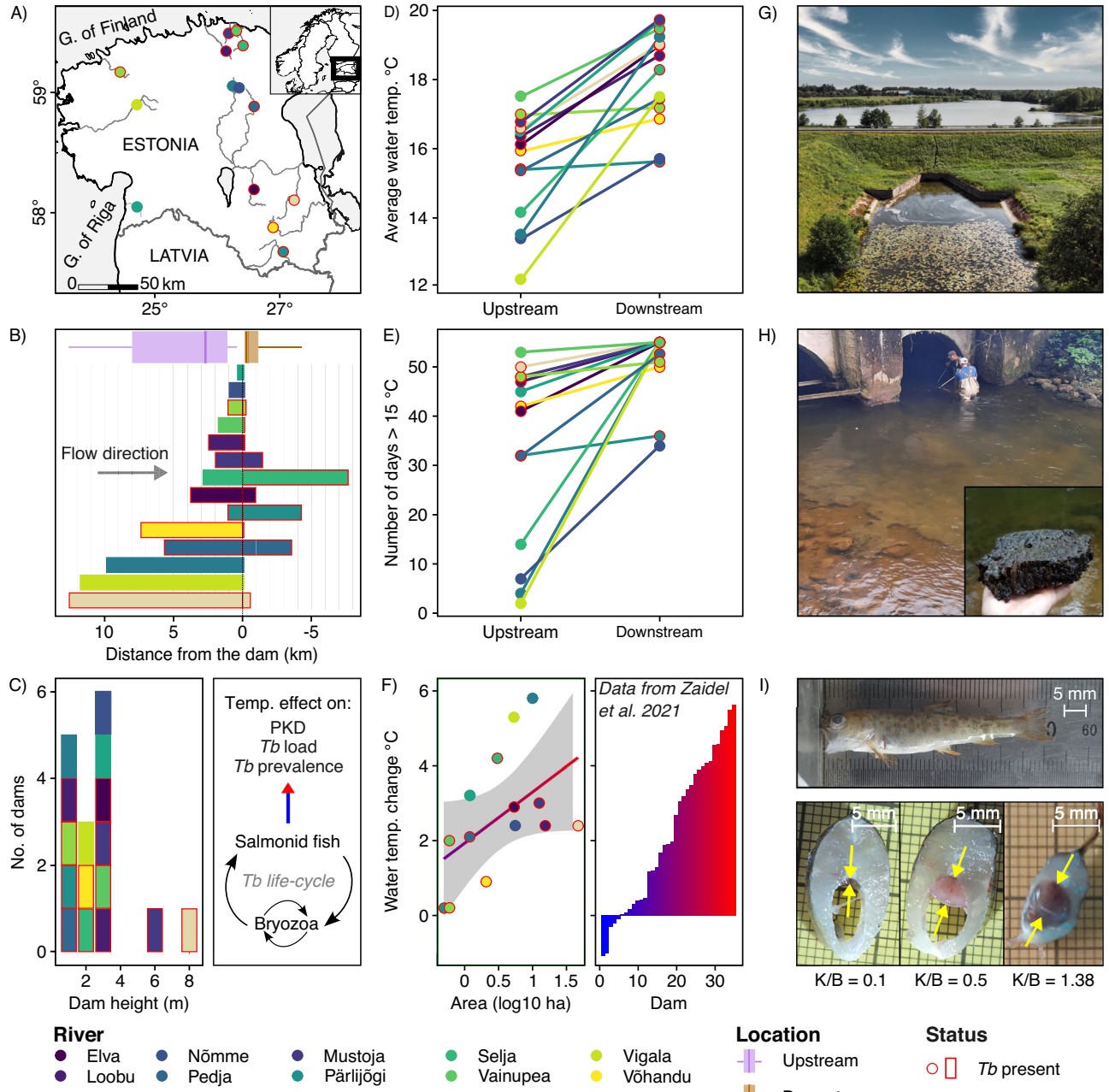

**Fig. 1 | Studied rivers and dams, water temperature and proliferative kidney disease (PKD) in brown trout. A** The study area consisting 14 dams. **B** Distances of the study sites from the dam, boxplots showing the distributions for up-and downstream sites. **C**. Left: Distribution of dam heights in relation to *Tb* occurrence including two consecutive dams in two rivers. Right: *Tb* life-cycle and scheme of the effect of temperature on parasite prevalence, load and PKD. **D** Mean summer water temperature with lines connecting upstream and downstream locations. **E** Number of days over 15 °C upstream and downstream of the dam. **F** Left: Effect of reservoir area (log10 ha) on water temperature change. Right: Temperature change downstream of 35 small dams according to Zaidel et al. (2021). **G** Aerial photo of small dam and reservoir (10.6 ha) at R. Mustoja. **H** Photo of a large bryozoan colony covering most of the bottom downstream of the dam at R. Ahja with a close-up image of the bryozoan colony (*Plumatella fungosa*). **I** Dead wild YOY brown trout found from downstream section of R. Mustoja with extreme renal hyperplasia. Sagittal cross section of a trout with normal (left), swollen (middle) kidney and extreme renal hyperplasia (right, dead trout from the same subfigure). The yellow arrows mark the location of the kidney, K/B indicates kidney-to-body thickness ratio as a measure of renal hyperplasia.

for extended periods of time.[19,24] We found that reservoirs significantly increase the duration of time with mean water temperature over 15 °C (on average with 18.1 days), which likely contributes to disease progression and increased mortality. Moreover, even very small reservoirs (1–3 ha) caused a rise in downstream water temperatures by several degrees. This highlights that, under low flow conditions, small reservoirs can substantially warm up the water, with potentially severe consequences on downstream biodiversity.[25,26]

## Dams give rise to parasite hotspots

We observed consistently higher infection prevalence and parasite load downstream of the dams. This is in line with both experimental- and field studies that demonstrate how increased water temperature results in higher *Tb* infection prevalence and parasite load in salmonids.[27,28] The temperature-driven proliferation of *Tb* within bryozoans and salmonids further enhances infectious *Tb* spore abundance in the environment.[17,19] Reservoirs also likely promote the growth, abundance and diversity of

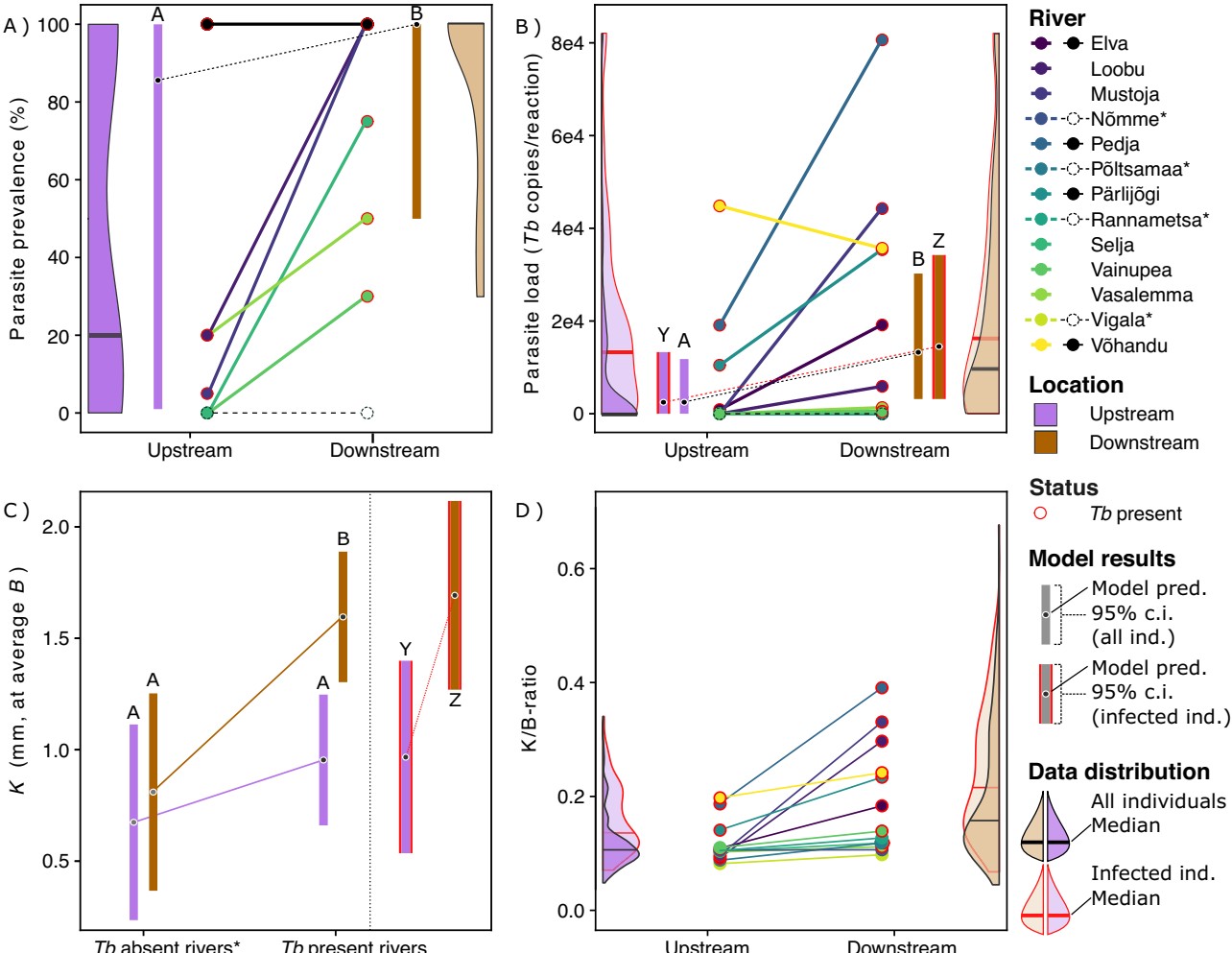

**Fig. 2 | Effect of dams on infection prevalence, parasite load and renal hyperplasia. A** Parasite prevalence (% of individuals infected) up- and downstream of the dams. Note that all rivers with 100% total prevalence (up- and downstream of dams) are shown with black colour and all rivers with 0% total prevalence are coloured white (see legend). **B** Parasite load (copies of *Tb* per reaction) up- and downstream of dams (upper clip at 8.2e4 for graphing purposes; see max values in Supplementary Fig. 3). **C** Relative kidney thickness (K) in relation to body thickness (dorsal muscular thickness, B), evaluated at the average B. **D** Relative kidney thickness measured as K/B-ratio (kidney thickness divided by dorsal muscular thickness). For **B**–**D** data and results are presented both for all trout individuals and for infected individuals only (see legend for colour scheme explanation of model results and data distribution). Significant differences are marked with different letters above model estimates (A-B for models including all individuals; Y-Z for models including only infected individuals).

### Dams as triggers of disease

Below the dams, brown trout exhibited higher levels of renal hyperplasia compared to their upstream counterparts. We also observed severe anemia, but only among individuals suffering from intense renal swelling. Elevated water temperature accelerates *Tb* proliferation in fish[20,28] and modulates the host's immune response.[36] At elevated temperatures, the fish immune system becomes overactive, leading to uncontrolled proliferation of interstitial kidney tissue, i.e. renal hyperplasia.[36] Acute PKD also reduces the concentration of red blood cells, limiting oxygen transportation capacity of the circulatory system and thereby reducing both aerobic scope and thermal tolerance of fish.[37] Thus, in addition to thermal stress, PKD-affected fish face a complex set of challenges, enduring reduced oxygen transport capacity, diminished aerobic scope, and compromised thermal tolerance,[37] which further challenges their ability to withstand elevated water temperatures.

Additionally, variable flow rates and increased evaporation from reservoirs affect fish through limiting habitat use.[38] During low-flow periods, salmonids often migrate to deeper pools and ponds,[39] which increases their risk of predation.[40] Reduced flow rates also elevate stress, increasing the risk of infection, as fish congregate in small areas leading to higher transmission rates.[14,41] Therefore, given widespread temperature- and density-

bryozoans, as in natural lakes.[29] Primary production is enhanced by increased water temperature, solar exposure and longer water residence times,[30] leading to a boost in food availability[31] for the primary host of *Tb*, bryozoans.[32] Hence, reservoirs and associated downstream river stretches likely become strongholds for bryozoans and *Tb*, as hinted by recent environmental DNA and niche modelling analyses.[33] The latter work illustrates the importance of water temperature, agricultural land use, sediment fineness and cumulative height of upstream dams in explaining *Tb* occurrence and abundance in brown trout. Therefore, our findings, together with existing ecological and parasitological knowledge[33–35], suggest that dams generate parasite hotspots for *Tb* downstream of reservoirs by creating favourable habitat for bryozoans. Future studies integrating eDNA data with hydrological and thermal profiles could help identify specific habitat features that facilitate bryozoan colonization. Longitudinal studies across seasons and dam types would further clarify how dam-induced habitat changes influence bryozoan abundances, parasite transmission dynamics and disease risk in salmonid populations. These effects may be indirect, via expanded bryozoan habitats that raise parasite spore abundance and infection risk, or direct, through higher disease prevalence and virulence driven by temperature-induced stress in fish.

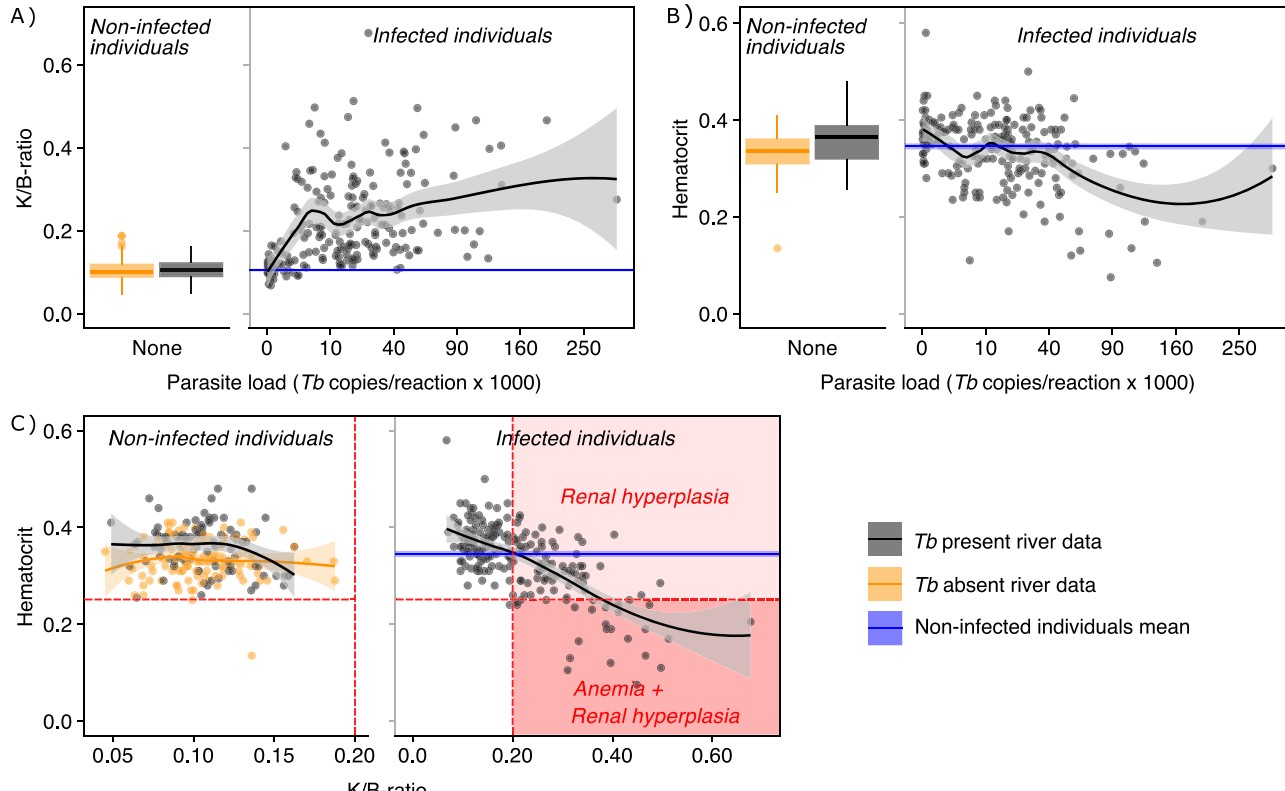

**Fig. 3 | Relationships between parasite load and PKD symptoms. A** Parasite load and renal hyperplasia (measured as K/B-ratio), **B** Parasite load and hematocrit, and **C** Renal hyperplasia and hematocrit, presented as loess regressions. Individuals from *Tb* absent rivers and uninfected individuals from *Tb* present rivers are treated as two sets of control data (all parasite loads = 0), presented as a Tukey boxplots to the left of the regressions, with their pooled mean values represented as a blue line (with 95% CI) crossing the regression graph.

dependence of many diseases,[42,43] reservoirs and dams that create migration barriers and thermal stress are likely to function as disease hotspots for other pathogens beyond PKD.

## Dams as potential barriers for parasite spread

In two studied rivers, we detected *Tb* in trout downstream of dams, while the parasite was not found upstream. Therefore, it is possible that these dams function as barriers for upstream parasite spread. Interestingly, in one of these rivers, the dam separating upstream and downstream location was removed in 2015 and functioning fish passage was built to allow anadromous trout to reach upstream spawning grounds. Seven years after construction of the fish passage, we did not observe *Tb* among juvenile trout upstream of the dam. This resembles a case in southern Germany where *Tb* spread has not been observed in 5 years after connectivity restoration.[44] Thus, while questions remain about the long-term risks of introducing parasites and invasive species to currently pathogen-free areas, restoring connectivity and natural lotic sections of the river is expected to provide substantial ecological benefits.[6,45,46] These include cooling down water, reduced algal growth, and improved access to cold-water refugia, ultimately benefiting a wide range of species. Furthermore, migration serves as a life-history strategy for salmonid fish to evade unfavourable parasite-rich environments, potentially leading to the evolution of pathogens with reduced virulence.[47,48] River temperatures also naturally rise from headwaters to downstream reaches due to cumulative effects of solar radiation, tributary inflows and decreasing elevation.[49] However, the measured water temperature increase (on average 2.64 °C) over the short distances involved (on average 5.5 km between sampling points) is greater than can be attributed to natural processes. Overall, while there is a potential risk of *Tb* spread to new upstream sections after dam removal, the environmental conditions will likely become less suitable for clinical PKD progression, and

ecological benefits may outweigh the associated risks. Equally important is restoring habitats in other heavily modified river sections to ensure connectivity, improved ecological functions and climate change resilience.[50] From the perspective of temperature-driven diseases such as PKD, another important measure is enhancing riverbank shading in artificially modified stretches that lack riparian vegetation, as this could significantly improve the temperature regime.[51]

## Limitations and implications

Although this study covers a single country, it provides important insights with potential implications beyond its borders. In Europe alone, the mean density of dam-like structures is estimated at 0.74 per river kilometre.[1] Along with habitat fragmentation, this is affecting temperature- and hydrological regimes of rivers. *Tb* infects a wide range of salmonid fishes (e.g. Atlantic and Pacific salmon, whitefish, char, grayling and trout) and is widely distributed in Europe and North America, with approximately half of the investigated salmonid populations being infected.[33,52–55] Recent outbreaks of PKD in North America,[56] coupled with an increase in disease prevalence in Europe, underscore PKD as an emerging climate-change-associated threat to salmonids.[16,57,58] Given that our analyses were based on surviving wild fish, the most severe disease cases were likely missed due to mortality, leading to an underrepresentation of the true impact of PKD.[59]

Due to the high abundance of dams in Estonia, our analyses included rivers with (*N* = 6) and without (*N* = 8) additional dams located upstream of our highest sampling sites. The average distance between the studied upstream study location and the next upstream dam was 6.1 km. Although upstream dams may have affected the measured variables, the paired design ensures robust results. Given the well-documented cumulative impacts of multiple reservoirs on water temperature and aquatic ecosystems,[60] we expect that multiple dams may have similar cumulative effects on disease dynamics.

Considering the ongoing threat of further fragmentation due to planned dam constructions,[61,62] our results are highly relevant for future efforts in hydropower development, dam removal, connectivity restoration, and riverscape rehabilitation. This urgency is further underscored by the rapid degradation of freshwater ecosystems[4] and the accelerating pace of global warming, which together highlight the need to better understand the interactive effects of disease and human-induced disturbances in aquatic ecosystems.[42,63]

## Materials and methods

### Experimental design

To investigate the impact of small dams on parasite infection dynamics and host pathology in wild brown trout, we employed a paired upstream-downstream sampling design. This approach allowed us to directly compare environmental and biological parameters across dam-influenced and non-impacted sites while minimising confounding spatial effects. We selected 14 river systems in Estonia and identified the nearest suitable upstream and downstream habitats for spawning and YOY brown trout. These paired sites were used to assess infection prevalence, parasite load and disease symptoms associated with *Tb*, the causative agent of PKD. Summer water temperature profiles were recorded using high-resolution loggers (Onset HOBO MX; HOBO Data Loggers, Bourne, MA, USA). Biological sampling and molecular diagnostics were then employed to link dam-induced thermal changes with parasite load and disease symptoms.

### Site selection and water temperature measurements

To estimate the effect of dams on infection prevalence, parasite load and PKD symptoms (renal hyperplasia, anemia), we sampled wild YOY brown trout from the closest spawning and recruitment areas immediately downstream and upstream of the dams to minimise the potential spatial effects (Fig. 1A, B). A total of 28 locations from 14 rivers containing man-made dams in Estonia were included to the analyses (Fig. 1A, and Supplementary Table 1). All studied dams and associated reservoirs were small (average dam height: 2.7 m, range: 0.95–7.65 m; average reservoir size: 8.1 ha, range: 0.5–48.5 ha; Fig. 1C, F, Supplementary Table 1). Eight of the studied dams have a functioning fish passage. Average study site distance from the dam was 0.9 km downstream and 4.6 km upstream of the dam (Fig. 1B, and Supplementary Table 1). Brown trout from five studied rivers were known to be infected with *Tb*,[64] whereas for the other nine rivers no prior parasite information was available. To measure daily water temperature, temperature data loggers were placed in each site at the end of May in 2022 (26 sites, 2 loggers per site) and at the end of June 2023 (2 sites). Water temperature data was measured every 6 h (06:00, 12:00, 18:00, and 00:00). For all studied rivers the mean summer water temperature was calculated for the period of June 29 to August 22 (55 days; longer timeseries available for 13 rivers). Based on maximum and minimum daily temperatures, we calculated diurnal variation for the same period of time (Supplementary Table 1, and Supplementary Fig. 1). Dam height and reservoir area data was obtained from the public Estonian Environmental Portal database (https://register.keskkonnaportaal.ee/). In two rivers (R. Mustoja and R. Loobu), two consecutive dams were present (distance between dams 0.65 and 3 km, respectively). Therefore, heights of 16 dams are presented in Fig. 1C and Supplementary Table 1. The area of the two consecutive reservoirs were summed when estimating the correlation between reservoir area and temperature change downstream of the dam (Fig. 1F, separate areas presented in Supplementary Table 1). Sampling of brown trout in those two rivers was carried out upstream of the upper dam and downstream of the lower dam. For comparative purpose, we extracted temperature data using https://plotdigitizer.com/app from a study which presents downstream mean water temperature change (August) for 35 small dams in Massachusetts, USA (Fig. 1F[12]).

### Sampling of fish

Standard electrofishing of YOY brown trout was conducted over 12-day period in Aug-Sept 2022 (14 rivers, 28 sites, permit no. 10-1/22/42-2 for 2022 and permit no. 10-1/23/50-2 for 2023, issued by Ministry of Regional Affairs and Agriculture of Estonia). In September 2023, we sampled one additional river (R. Loobu) and repeated electrofishing in R. Ahja where YOY brown trout were not detected downstream of the dam in 2022. However, similar to 2022, YOY trout were not detected in 2023, although several older trout specimens with detected *Tb* infection were caught in both years. Therefore, R. Ahja was not included to parasitological analyses, and 26 sites located upstream and downstream of dams in 13 rivers were analysed further. Previous studies have shown that *Tb* prevalence in infected salmonid populations is typically >0.2.[52,53,65] Therefore, we aimed to collect tissue samples from ca 20 YOY brown trout per location (exact sample sizes are presented in Supplementary Table 1) to minimise the number of lethally sampled fish, while maintaining sufficient power (95%) to distinguish non-infected and infected populations when true prevalence is >0.2.[66] The experimental unit in this study was an individual fish, each measured separately. However, fish collected in the study are not independent within river sections, which is handled by implementing mixed models including location as a random factor. Caught YOY brown trout were held in an aerated container, following immediate sampling to minimise stress and confounding effects. No expected or unexpected adverse events were observed during capture, handling, or euthanasia of the fish. Fish were sampled in random order from the holding tank, providing no systematic bias related to capture sequence. Each specimen was euthanized with an overdose of benzocaine (>250 mg L$^{-1}$, Caesar & Loretz GmbH, Hilden, Germany) in accordance to the principles described in the EU Directive 2010/63/EU, on the protection of animals used for scientific purpose. No additional ethics approval was required for the collection of brown trout in Estonia, as the capture of wild fish is regulated under the local Animal Protection Act (https://www.riigiteataja.ee/en/eli/ee/521032019002/consolide/current), all brown trout were legally caught, and the species is not under protection in Estonia. This investigation was conducted as a field study involving naturally occurring brown trout populations, and therefore did not involve experimental manipulation. We have complied with all relevant ethical regulations for animal use.

Fork length was measured using a measuring board to the closest 0.5 mm. Total mass (mg) of each fish was measured with a precision scale (Radwag WLC 2/A2/C/2; RADWAG, Radom, Poland). Blood samples were collected from the caudal artery using heparinized microcapillary tubes (0.5–0.6 mm in diameter; Paul Marienfeld GmbH & Co.KG, Lauda-Königshofen, Germany) and centrifuged immediately after sampling at 12,250 × $g$ for 5 min (QBC Capillary Centrifuge; Drucker Diagnostics, Port Matilda, PA, USA). Blood plasma and packed red blood cell estimates were carried out using a standard ruler (to closest 0.5 mm; [67]). One, two or three microcapillaries were collected per fish from 427, 222 and 6 individuals, respectively, and mean hematocrit values per fish was calculated when multiple estimates were available. In order to quantify renal hyperplasia, a sagittal cross-section of each fish was produced by cutting the body between the front tip and the end of the dorsal fin.[37] This section was photographed against 1 mm grid (Fig. 1I) with a digital camera for the measurement of the kidney-to-body thickness ratio (K/B-ratio) as a quantitative estimate of renal hyperplasia using ImageJ software.[37,68] The cross-section was stored in 96% ethanol (5 mL screw-cap centrifuge tubes) for DNA extraction from the kidney tissue for quantification of *Tb*.

### DNA extraction and molecular quantification of the parasite

Total genomic DNA from kidney tissues of the 446 YOY brown trout were extracted using QIAamp 96 DNA QIAcube HT kit and the QIAcube® HT Instrument for automated nucleic acid purification (QIAGEN, Hilden, Germany). Each 96-well extraction plate included one negative DNA extraction control sample. DNA concentrations were measured with NanoDrop 2000 (Thermo Fisher Scientific, Waltham, MA, USA) and diluted to 20 ng μL$^{-1}$. To reduce variation associated with spectrophotometric DNA quantification, DNA dilutions were subsequently re-measured and adjusted if the estimated values deviated by more than 2 ng μL$^{-1}$ from the target concentration. The quantification of *Tb* was

performed in a set of three replicates for each kidney extraction using qPCR (quantitative polymerase chain reaction) based on Taqman probe on a LightCycler 480 (Roche, Basel, Switzerland). The assay employed *Tb* specific primers (PKX18s 1337f: 5′-CGAACGAGACTTCTTCCTT-3′, PKX18s 1426r: 5′-CTTCCTACGCTTTTAAATAGCG-3) and TaqMan hydrolysis probe PKX18s 1399p (5′-FAM-CCCTTCAATTAGTTGATC-TAAACCCCAATT-iQ500-BHQ-1-3′) to amplify a 90-bp 18S rDNA sequence of the parasite.[69] Each 10 µL amplification reaction consisted of 4.4 µL of nuclease-free water, 2 µL of 5× HOT FIREPol Multiplex qPCR Mix (NO-ROX; Solis BioDyne OÜ, Tartu, Estonia), 0.2 µL of 200 nM forward and reverse primers, 0.2 µL of 200 nM probe, and 3 µL of genomic kidney DNA (20 ng µL$^{-1}$, total of 60 ng of total tissue DNA per reaction). qPCR was conducted using the following thermal cycling conditions: an initial denaturation step at 95 °C for 10 min, followed by 45 cycles of 95 °C for 15 s and 60 °C for 60 s. All qPCR plates were manually prepared and run by the same individual.

A tenfold serial dilution of pooled *Tb*-positive kidney DNA, with total genomic DNA concentrations ranging from 40 to 0.004 ng µL$^{-1}$ (five concentrations), was prepared and included on each plate in six technical replicates per concentration as a standard dilution series. Negative PCR controls were included on each plate in six replicates (36 total negative PCR controls). None of the negative PCR controls showed positive amplification signal. Additionally, one negative DNA extraction control was included on each qPCR plate (six plates total), with three technical replicates per plate. None of the negative DNA extraction controls showed positive amplification.

To estimate the Limit of Detection (LOD) and Limit of Quantification (LOQ), a synthetic dilution series was created by using an artificially synthesized 90 bp *Tb* 18S gene sequence fragment.[69] The series began with a tenfold serial dilution ranging from 6,022,142 to 60.2 copies µL$^{-1}$ (five concentrations, with 32 technical replicates per concentration). This was followed by an eightfold dilution, reducing the concentration from 60.2 to 7.5 copies µL$^{-1}$ (32 technical replicates per concentration). Finally, two fivefold dilutions were performed, further decreasing the concentration from 7.5 to 1.5 and then to 0.3 copies µL$^{-1}$ (64 technical replicates per concentration).

Due to the substantial variation in parasite load among infected samples (e.g. up to $1.6 \times 10^6$ orders of magnitude[70]), an SD threshold of <1 was used to identify outliers. Based on this threshold, two samples were excluded from the dataset because removing a single technical replicate did not reduce the SD to <1. Additionally, a single technical replicate was removed from five samples to achieve an SD of <1 (samples included in the dataset). After outlier removal, the mean SD across all samples was 0.11 (Supplementary Table 10). Regression analysis of the standard dilution series was performed for each plate to determine qPCR plate efficiencies (Supplementary Table 10). LOD (3.6 copies/reaction) and LOQ (9 copies/reaction) were determined using the synthetic *Tb* 18S rRNA gene dilution series.[71,72] The effective LOD for three replicates was calculated as 1.1 *Tb* target copies/reaction.

## Statistics and reproducibility

All analyses were run in R 4.4.2 (R Core Team, Vienna; https://www.r-project.org). For statistical comparisons, we ran LMMs, or generalized linear mixed models (GLMM) using the *lme4*[73] and *lmerTest*[74] packages (presented below in *lme4* syntax). Significance of included factors was evaluated using the Wald $\chi^2$-test through the *car* package.[75] Graphs were constructed based on elements created in *ggplot2*[76] and *cowplot*[77] and formatted in Inkscape 1.3.2 (Inkscape Project 2020, https://inkscape.org/). Statistical summary tables from raw data and models are presented in the supplementary materials (Supplementary Tables 2–10).

Individuals were excluded from a given analysis only when the relevant measurement was unavailable, no other a priori exclusion criteria were defined. Specifically, two fish lacked total length measurements, one lacked kidney/body thickness values, and 18 lacked hematocrit measurements. At the river level, R. Ahja was excluded from morphological and disease-trait analyses due to missing data. Final sample sizes reported for each analysis reflect these exclusions.

The non-parametric Wilcoxon signed-rank test was used to compare water temperature, days with water temperature >15 °C and diurnal water temperature fluctuation in up- and downstream locations of the dams and reservoirs ($N = 14$ rivers, 28 sites).

Body size differences ($N = 442$ individuals) between upstream and downstream locations ($N = 13$ rivers, 26 sites) were assessed based on *total body length* (tl), using the LMM: tl ~location + (1|river), where location is a fixed factor (two levels: upstream- and downstream of dam) and river is a random factor [13 levels (river names)]. Body condition between upstream and downstream locations was evaluated as the *relative body mass* (m) given the total length of the individuals; model: $\log_e(m) \sim \log_e(tl) + location + (1|river)$. Model assumptions related to residual normality and homoscedasticity were evaluated using QQ-plots and residuals vs. fitted plots, and were judged to be acceptable in the presented models.

Infection prevalence (proportion of infected individuals; pv, ($N = 13$ rivers, 26 sites) was analysed with the GLMM: pv ~ location + (1|river), weights = tot.obs, family = binomial. For weights, we used the total number of individuals (tot.obs) that each location-specific proportion value was based on. Overdispersion was tested through the *DHARMa* package[78] and found non-significant (dispersion = 1.14, $P = 0.30$). The same model was run using a data subset including only rivers where the parasite was detected ($N = 9$ rivers, 18 sites), given that effects are mainly of interest in rivers with confirmed parasite presence. Inflation of expected values was specifically investigated by comparing models with and without random effects and simulated predictions. Random effect variance was high, resulting in inflated expected values. Hence, the GLMM was also followed by a simpler pairwise *t*-test to investigate conformity of results.

Parasite load (pl) was square-root transformed to accommodate for positive skew and analysed with the LMM: sqrt(pl) ~ location + (1|river); focusing on comparing upstream and downstream locations overall ($N = 444$ individuals, 13 rivers). Residual diagnostics (QQ-plot and residuals vs. fitted plots) were deemed acceptable for general interpretation of significant differences after transformation. We also ran the same model using a data subset including only rivers where the parasite was detected ($N = 315$ individuals, 9 rivers).

Renal hyperplasia was analysed as the relative thickness of the kidney to dorsal muscular tissue (Fig. 1I). In models, *kidney thickness* (K) was the dependent variable, height of dorsal musculature (*body thickness*; B) was added as a covariate [LMM: K ~ B + location + (1|river)] representing a proxy of the overall size of the fish; in data plots, we use the calculated K/B-ratio ('K/B-ratio' = K/B, $N = 443$ individuals). Residual diagnostics /QQ-plot and residuals vs. fitted plots), indicated higher variation in downstream sites (heteroscedasticity at the location level), but the model was deemed acceptable for interpretation of general differences. A follow-up model was built utilizing the fact that some rivers ($N = 4$) were completely parasite-free in the analysed samples (ad hoc controls), adding the factor *parasite presence* (par.pres) in interaction with location [model: K ~ B + location × par.pres + (1|river)]. Results from the latter model were analysed using pairwise location × par.pres contrasts (Kenward–Roger method for degrees-of-freedom; Tukey adjustment for multiple comparisons), using the *emmeans* package.[79]

Progression of renal hyperplasia (quantified as K/B-ratio) in relation to parasite load was analysed for infected individuals ($N = 213$ individuals), using a non-linear loess regression (pooling all rivers). The relationship between the variables was assessed graphically based on the 95% confidence band relative to the estimated mean K/B-ratio for uninfected individuals (non-overlapping confidence bands indicating significant difference, $N = 230$ individuals).

Hematocrit was analysed in relation to parasite load and renal hyperplasia using non-linear loess regressions, as above ($N = 209$ infected & 217 uninfected individuals). Reservoir area and downstream temperature rise was assessed using linear regression ($N = 14$ reservoirs).

## Inclusion & ethics statement
Field sampling was conducted under relevant permits and followed national and institutional animal welfare guidelines. The author team includes researchers from diverse backgrounds, career stages and geographic locations.

## Reporting summary
Further information on research design is available in the Nature Portfolio Reporting Summary linked to this article.

## Data availability
All data and code supporting the findings of this study are available in the Figshare repository under the https://doi.org/10.6084/m9.figshare.28595468.v4. Source data for all figures and analyses can be accessed directly through this repository. Any additional information is available from the corresponding author upon reasonable request.

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

## Acknowledgements

We thank Alfonso Diaz Suarez, Gustav Lauringson, Karl-Erik Aavik, Tanel Ader, Herki Tuus and Veljo Kisand for their assistance and support, Põlula Fish Rearing Centre of State Forest Management Centre (RMK) for logistical support and Oksana Burimski for laboratory assistance. Vihula III dam and reservoir photo (10.6 ha) at the Mustoja river, shown in Fig. 1G, is based on a drone photo from 10.07.2023, kindly provided by Gunnar Laak. The study was funded by Estonian Ministry of Regional Affairs and Agriculture, contract no. 4-1/22/6, Estonian Research Council, grant no. PSG849, PRG852, PRG3078 and Swedish research council for sustainable development FORMAS, grant no. 2021-01643.

## Author contributions

A.V. and M.L. contributed equally to this work. A.V. and M.L. conceived the study. Methodology was developed by A.V., J.N., L.P., M.L. Investigation was carried out by A.V., L.P., M.L., R.G. and S.K. Visualization was performed by A.V., J.N. and M.L. The original draft was written by A.V., J.N. and M.L., and all authors contributed to review and editing.

## Funding

## Competing interests

The authors declare no competing interests.
