## [Transparent Peer Review file · Communications Biology]

Dams threaten salmonids by triggering temperature-dependent proliferative kidney disease

Corresponding Author: Professor Anti Vasemägi

Version 0:

Decision Letter:

**** Please ensure you delete the link to your author homepage in this email if you wish to forward it to your coauthors ****

Dear Professor Vasemägi,

Your manuscript entitled "Dams threaten salmonids by triggering temperature-dependent disease" has now been seen by 3 referees, whose comments are appended below. You will see from their comments below that while they find your work of considerable interest, some important points are raised. We are interested in the possibility of publishing your study in Communications Biology, but would like to consider your response to these concerns in the form of a revised manuscript before we make a final decision on publication.

We therefore invite you to revise and resubmit your manuscript, taking into account the points raised. In particular, please note that the following revisions would be necessary for us to contact our referees again:

1 – Please address the points 106, 107, 142 and 167 from reviewer #1.

2 – Please address the concerns from reviewer #2 about the measurement of Tb loads and the response variable.

Additional comment from the editor I: Please note that reviewer #2 has provided additional feedback in the attached pdf file.

Additional comment from the editor II: Please note that it is not mandatory to include additional data about bryozoans (intermediate host) as suggested by reviewer #3 (if these data have not been collected already).

We are committed to providing a fair and constructive peer-review process. Do not hesitate to contact us if you wish to discuss the revision in more detail or if there are specific requests from the reviewers that you believe are technically impossible or unlikely to yield a meaningful outcome.

At the same time, we ask that you ensure your manuscript complies with our editorial policies. Specifically:

For all graphs depicting a single point value (e.g., mean) with error bars, you must add individual data points or convert the graph to a boxplot or dot-plot to show data distribution.

It's mandatory to provide access to the numerical source data for graphs and charts either through a repository or by providing the data in a Supplementary Data file (in excel format).

All blots/gels must be accompanied by size markers in every figure panel. Uncropped and unedited blot/gel images must be included as Supplementary Figure(s) in the Supplementary Information pdf.

Please ensure that you have complied with the data deposition policies at the Nature Portfolio, please see here.

Please ensure that you have complied with our policies on research involving animals and humans, see here

Please follow the ARRIVE guidelines for reporting animal experiments. Please fully complete an [ARRIVE checklist](https://arriveguidelines.org/sites/arrive/files/documents/Author%20Checklist%20-%20Full.pdf) including both the essential and recommended set of items (adding information to the manuscript where needed) and upload this with your revised manuscript.

Please also see [our revision checklist](https://www.nature.com/documents/CommsBio-file-checklist-revision.pdf) for guidance on formatting the manuscript and complying with our policies. A comprehensive guide to our formatting requirements for final submissions is also available for your reference [here](https://www.nature.com/documents/commsj-life-style-formatting-guide-accept.pdf).

Please use the following link to submit your revised manuscript, point-by-point response to the referees' comments (which should be in a separate document to the cover letter) and any additional files:

Link Redacted

When submitting the revised version of your manuscript, please pay close attention to our [Digital Image Integrity Guidelines](https://www.nature.com/commsbio/editorial-policies/image-integrity).

We would like to receive your revision within 4 weeks, but appreciate that every situation is unique. We look forward to receiving your revised manuscript when it is ready, and will not enforce a hard deadline on this revision.

Please do not hesitate to contact me if you have any questions or would like to discuss these revisions further. We look forward to seeing the revised manuscript and thank you for the opportunity to review your work.

Best regards,

Johannes Stortz, PhD
Senior Editor
Communications Biology
orcid.org/0000-0002-5928-1850

on behalf of

Eoin O'Gorman, PhD
Editorial Board Member
Communications Biology
orcid.org/0000-0003-4507-5690

Reviewers' comments:

Reviewer #1 (Remarks to the Author):

In this important manuscript, the authors convincingly show how dams in rivers may exacerbate a deadly salmonid kidney disease. This is likely due to a combination of the exposure of stagnant water in the reservoir to solar radiation increasing the temperature more than 2 °C and possibly favourable conditions in the reservoir for bryozoan growth, which are the primary hosts for the parasite causing the kidney disease. The data combine a large study of 14 rivers with dams showing an increase in parasite prevalence, kidney disease, and anaemia from upstream to downstream of the dam. Apart of bringing awareness of the negative effects of damming for sensitive trout populations, the data also show the negative impact on valuable natural trout populations of a temperature rise that must be expected in the near future when global warming is not brought to a halt. The paper thus has both scientific value and interest for the broader public and informs decisions about river management and discussions about future sustainability.

The paper is well written and analysed and I have only minor comments. Mainly I feel that the results section could be optimized, by using more consistency in annotations, and by clarifying the statistical results. Also, please clarify the N-values a bit better (number of rivers or number of sample sites?). These points I have left ordered by line number:

95: Please be consistent with annotations: studied cases, dams, reservoirs, rivers are all the same so only use one not to confuse the reader.

97: Consistency: similar for location, which is used in the statistical analysis for upstream vs downstream. 98 area should be

location. Do not use location as annotation for something else later! I think the whole problem is the use of location and locations where with locations you mean the river stretch in which you sampled. That should be double the number of rivers. in the methods you mention sites: line 312. Anyway, this is an annotation problem you should solve and then keep using it consistently!

100: missing U value and N-value for Wilcoxon test

101: change "no difference" to "no significant difference"

104: the beta is positive but the CI values are negative. I guess you should make them positive.

105: change to: Sampled brown trout did not differ significantly in body size ... (N = 14 but here N = 13, explain why)

106: delete ", neither in total length"?

106: rather than giving ANODEV as information here about the test I would find it more helpful to give LMM as information (or GLMM). The ANODEV of car is a method to get to a statistical evaluation of that test, in this case an ANOVA F-test would be an appropriate solution for the linear approach (but the Wald test you use does fine as well).

106: not so important for such a low chisquare, still what is df for this test?

107: As a query for body condition you regressed log(mass) on log(total length) as given in 391. This apparently gives a beta difference for the effect of location of -0.022 (LMM is the test and could be handled as the other test before to keep the annotations similar). You write that body condition is slightly lower? This is however difficult to envision from the beta you give in your model. What exactly means the -0.022? if this means that the relationship between log(w) and log(tl) changes than you have a change in the allometric relationship $w = a * tl^b$, where b is commonly around a value of 3. That would be a less steep relationship in the upstream fish? Anyway it gets a bit confusing and I wonder why not use Fulton's condition factor? Then you could show how much that factor has changed from upstream to downstream.

112: Here you present the data of two analysis in which the second one is on a subset of the data only including rivers in which the Tetracapsuloides was detected. This takes some effort for the reader to understand. Better would be to show only one of the two, and I find the second one more relevant. The other can remain in the supplement. Then first write that xx of xx rivers were found positive for Tetracapsuloides briosalmonae and then do the rest of the analysis only with this subset.

112-113: the use of locations is confusing here. In the model, you use river (equivalent for dam (with a total N of 14)) as random factor. So should this not be N = 13 and N = 9 rivers? In this respect note that you write "Hence, river level variability has a dominant influence on model predictions" in 115!

113: Tb-present is a complex annotation. If you first give how many rivers were tested positive you can introduce the annotation. Is that 9 of 14 rivers / dams?

118: delete "simple" and write that you used mean values per river / location

119: How do N = 18 locations result in df = 12 in a paired t-test? A df of 12 indicates you had 13 paired values which includes all the rivers (14) -1.

124: Not clear what the beta would tell me at this point. In case searching for additional information you could consider effect size. Further report which test you used LMM or GLMM and the model p-values etc.

132: Here as for condition you regressed the values. That makes it complex to read the results. Why not test the K/B ratio? Further Kidney hyperplasia is a condition where K/B ratio is high. Consider rewriting this part.

137: would you still call it kidney hyperplasia here? Better refer to K/B ratio.

137: You mean that the fish in downstream locations of Tb absent rivers have lower K/B ratio than fish in ... Or K/B ratio was lower... etc.

142: What is the definition you use of low parasite loads. Better write descriptive and give the numbers and ranges you found. Similar for 147 Hematocrit

167: This is the first time you mention a focus on "brown trout recruitment areas" and as is raises confusion about what you mean with this or how it could affect your measurements. This is further only mentioned in the method section. Best would be to add it to the end of the introduction and give some table where we can see how far up and downstream of the dam the data were collected. Moreover, in 232 you correctly mention that waters may naturally warm up in summer while flowing downstream. However, I assume you do not want this to downgrade your central argumentation that it is the damming that increases the water temperature. Now you mention it please argue how this may not explain the extent of increase in temperature to keep your central argumentation intact.

188: delete "(class Phylactolaemata)"

229: There is some new paper on cool water habitat choice in brown trout in relation to PKD, maybe cite that...

261: The 1C figure does not show: Left: Distribution of dam heights in relation to Tb occurrence. Why are there 16 dams (coloured blocks) in this figure? Also not clear what is the function of the right panel of 1C.

285: and figure 1B, is it possible to calculate a relationship between the distance between the two measure points and temperature change with reservoir size as random variable? How does this value stand in relation to the estimated temperature increase due to the reservoir itself?

400: The square-root transformation should not be between brackets.

700: In the figure you write renal hyperplasia (also in methods line 328) but in the main text kidney hyperplasia. Keep things consistent...

Reviewer #2 (Remarks to the Author):

Dear Authors

It was a pleasure to read your MS. It is an interesting and very well-written MS. It tackles a novel and pressing question based on a solid and clever design. I don't have much major comments; all is clear both the data and analyses are strong. My comments are directly embedded into the attached pdf file. Most of these comments can be considered as "minors" but two: (i) the way Tb load is measured (qPCR from kidney DNA) is fine but I would like to have a bit more information about whether it is problematic or not to not correct for trout DNA when estimating TB load, and (ii) this response variable might be better modeled using quasipoisson (or negative-binomial) error terms. See the comments in the file for details and more explanation.

Very nice job, congratulations

Best wishes

Reviewer #3 (Remarks to the Author):

Comments to authors

This study presents a set of very clear results associating dams to infection risk and virulence of *Tetracapsuloides bryosalmonae*. Association with the PKD disease and temperature is well known but authors are the first to point out that small reservoirs warm up the water and this alone may lead to increase in disease prevalence and virulence. This result is not surprising for those who work in the field but as the data are very convincing this is a very important result for management and restoration of watersheds. It is an especially relevant feedback to the ongoing discussion of the importance of small hydropower plants for sustainable energy production in Europe. Here the message is clear, small hydropower will come with potential disease problems in salmonid fish. It would be very important to have these data in the discussion of watershed management and restoration.

I have largely only positive comments on the actual study design, data analysis and interpretation. The design is a solid pairwise upstream-downstream design with sufficient replication. The key result seems large and robust. There is no reason to doubt that these results would not be more general across the PKD / trout range. Methods to detect the parasite, measure the effects of parasite on fish and record the necessary environmental parameters are state of the art and reliable. Statistical analysis is appropriate, but also the effect sizes and the key response seem very clear and robust. Writing is clear and figures are already standalone giving a clear overview of the results.

I find this study valuable and well-conducted.

I have one comment that authors might want to consider. This comment is about direct and indirect effects of temperature increase due to dams. Authors would have the opportunity to deepen the conceptualization of the study by discussing the direct and indirect effects more clearly. I am sure they actually think like that, but it would be good to spell it out.

I try to explain what I mean. Readers who know the system will be wondering if the temperature increase benefits the bryozoans that are the intermediate host of the myxozoan parasites. Authors devote a paragraph in the discussion to the possible increase in bryozoan population due to changes in the flow regime and temperature. I think it would be useful to bring the thematic of multiple host life cycle already earlier, even mentioning that in the abstract. One could explain that with multiple host life cycle processes in the intermediate host (indirect temperature effects, for example) may be important. Why do I think this would be important? This would allow authors to explain how direct and indirect effects of increasing temperature operate in this system. PKD is an example of a parasite which has higher virulence in stressed host (trout), and here authors measure virulence. Temperature will increase the stress levels of fish. PKD is also an example of a parasite where parasite risk depends on density of infected bryozoans in the environment. Bryozoans are filter feeders that thrive in

rivers with pools of slowly flowing water, solid (built) structures and low water level variation. Bryozoans probably like impoundments like authors write. If there is something missing in the study it is the assessment of the density of infected and uninfected bryozoans upstream, in reservoirs and downstream. It would be a great addition to the study, but probably authors did not sample bryozoans. I might be asking a bit too much, but it would be great to be able to dissect / discuss the direct and indirect effects of the temperature. Direct effects would be the increase in disease prevalence and virulence due to stressed fish (temperature dependent susceptibility and virulence). Indirect effects would be the epidemiological consequences of more favorable bryozoan habitats (increase in parasite spore numbers and infection risk). This distinction might have importance for management decisions. Authors could also write the introduction and motivation of the study by explaining the expectations with respect to direct and indirect effects of temperature, showing that the problematic with abiotic stressors comes with indirect effects through ecological change in the environment.

** See the Nature Portfolio author and referees' website at www.nature.com/authors for information about policies, services and author benefits

Communications Biology is committed to improving transparency in authorship. As part of our efforts in this direction, we are now requesting that all authors identified as 'corresponding author' create and link their Open Researcher and Contributor Identifier (ORCID) with their account on the Manuscript Tracking System prior to acceptance. ORCID helps the scientific community achieve unambiguous attribution of all scholarly contributions. You can create and link your ORCID from the home page of the Manuscript Tracking System by clicking on 'Modify my Springer Nature account' and following the instructions in the link below. Please also inform all co-authors that they can add their ORCIDs to their accounts and that they must do so prior to acceptance.

Version 1:

Decision Letter:

** Please ensure you delete the link to your author homepage in this email if you wish to forward it to your coauthors **

Dear Professor Vasemägi,

Your manuscript entitled "Dams threaten salmonids by triggering temperature-dependent disease" has now been seen again by our referees, whose comments appear below. In light of their advice I am delighted to say that we are happy, in principle, to publish a suitably revised version in Communications Biology.

We therefore invite you to edit your manuscript to comply with our format requirements and to maximise the accessibility and therefore the impact of your work. Please note that addressing the remaining points from reviewer #2 is not mandatory.

* Please see the attached document for editorial requests for the final version (.docx file). Please ensure a completed version of this file is uploaded as a Related Manuscript with your final submission.

* Please review our [final submission file checklist](https://www.nature.com/documents/commsj-file-checklist.pdf) to ensure all necessary files are present with your final submission and to avoid delays in accepting your manuscript. For your reference, a style and formatting guide is available [here](https://www.nature.com/documents/commsj-life-style-formatting-guide-accept.pdf) and includes all of our style requirements.

It is important that you pay careful attention to the requests in these documents to avoid a delay in formal acceptance of the article.

Open access

Communications Biology is a fully open access journal. Articles are made freely accessible on publication. For further information about article processing charges, open access funding, and advice and support from Nature Research, please visit <https://www.nature.com/commsbio/open-access>

At acceptance, you will be provided with instructions for completing the open access licence agreement on behalf of all authors. This grants us the necessary permissions to publish your paper. Additionally, you will be asked to declare that all

required third party permissions have been obtained, and to provide billing information in order to pay the article-processing charge (APC).

Please use the following link to upload your revised files:

Link Redacted

We hope to hear from you within two weeks. If you expect the process to take longer than one month, please let us know.

Congratulations on an excellent paper!

Best regards,

Johannes Stortz, PhD
Senior Editor
Communications Biology
orcid.org/0000-0002-5928-1850

Eoin O'Gorman, PhD
Editorial Board Member
Communications Biology
orcid.org/0000-0003-4507-5690

PS: At acceptance, the corresponding author will be provided with instructions for completing the license on behalf of all authors. This grants us the necessary permissions to publish your paper. Additionally, you will be asked to declare that all required third party permissions have been obtained, and to provide billing information in order to pay the article-processing charge (APC).

REVIEWERS' COMMENTS:

Reviewer #1 (Remarks to the Author):

All my queries have been satisfactorily answered by the authors. The result is an excellent manuscript that deals with important data on the impact of dams on diseases in a warming world. I have no further comments.

Reviewer #2 (Remarks to the Author):

Dear Authors

Thanks for the corrections you've made on the MS. It seems you have replied to all the comments made by the reviewers rather adequately. I still think that quasipoisson/negbin models are better than Gaussian models when we are to work with count data, but given the strong effects reported by the authors, I don't think choosing one or the other would change the conclusions. Nonetheless, the authors could have ran (just as an information for the reviewers) a negbin or quasipoisson GLMM; this is super straightforward (glmmPQL for quasipoisson, glmmTMB for negbin) and permit reinforce further the conclusions.

Apart from that I'm fine with the MS as it is.

Reviewer #3 (Remarks to the Author):

Thank you for considering my suggestions and incorporating a bit more discussion of indirect and direct effects of temperature in the discussion. I think the manuscript is very nice and response of the authors sufficient and considerate.

** See the Nature Portfolio author and referees' website at www.nature.com/authors for information about policies, services and author benefits

We would like to thank editors and the reviewers for the time and effort spent evaluating our manuscript entitled “Dams threaten salmonids by triggering temperature-dependent disease” (COMMSBIO-25-5900-T) by Magnus Lauringson, Joacim Näslund, Lilian Pukk, Siim Kahar, Riho Gross and Anti Vasemägi. We appreciate the constructive feedback and have carefully revised the manuscript in response to the comments provided. Below, we address each point raised by the editors and reviewers and explain the corresponding changes made.

	Reviewer #1 (Remarks to the Author): In this important manuscript, the authors convincingly show how dams in rivers may exacerbate a deadly salmonid kidney disease. This is likely due to a combination of the exposure of stagnant water in the reservoir to solar radiation increasing the temperature more than 2 °C and possibly favourable conditions in the reservoir for bryozoan growth, which are the primary hosts for the parasite causing the kidney disease. The data combine a large study of 14 rivers with dams showing an increase in parasite prevalence, kidney disease, and anaemia from upstream to downstream of the dam. Apart of bringing awareness of the negative effects of damming for sensitive trout populations, the data also show the negative impact on valuable natural trout populations of a temperature rise that must be expected in the near future when global warming is not brought to a halt. The paper thus has both scientific value and interest for the broader public and informs decisions about river management and discussions about future sustainability. The paper is well written and analysed and I have only minor comments. Mainly I feel that the results section could be optimized, by using more consistency in annotations, and by clarifying the statistical results. Also, please clarify the N-values a bit better (number of rivers or number of sample sites?). These points I have left ordered by line number:	We thank the reviewer for the thorough review and positive feedback. We have now clarified the use on N throughout the manuscript.
1.	95: Please be consistent with annotations: studied cases, dams, reservoirs, rivers are all the same so only use one not to confuse the reader.	Terminology for sampling units has been harmonized. We now use “location(s)” to describe the downstream and upstream sampling sites and “rivers” as a general unit based on analysis of several rivers including their respective sampling locations.
2.	97: Consistency: similar for location, which is used in the statistical analysis for upstream vs downstream. 98 area should be location. Do not use location as annotation for something else later! I think the whole problem is the use	Terminology for sampling units has been harmonized. We now use “location” to describe a single sampling site, in plural both up- and downstream sites and “river(s)” as a general unit based on analysis of both locations.

	of location and locations where with locations you mean the river stretch in which you sampled. That should be double the number of rivers. in the methods you mention sites: line 312. Anyway, this is an annotation problem you should solve and then keep using it consistently!	
3.	100: missing U value and N-value for Wilcoxon test.	We have now added missing df, U and N values for Wilcoxon tests.
4.	101: change "no difference" to "no significant difference".	Corrected to "no significant difference", line 118 - 119.
5.	104: the beta is positive but the CI values are negative. I guess you should make them positive.	Corrected.
6.	105: change to: Sampled brown trout did not differ significantly in body size ... (N = 14 but here N = 13, explain why).	The variation in sample size is clarified under Materials & Methods section "Sampling of fish", as we did not catch any YOY trout downstream of one dam (R. Ahja, Saesaare dam) in two consecutive years. We have clarified this also under the current section (lines 123 to 125).
7.	106: rather than giving ANODEV as information here about the test I would find it more helpful to give LMM as information (or GLMM). The ANODEV of car is a method to get to a statistical evaluation of that test, in this case an ANOVA F-test would be an appropriate solution for the linear approach (but the Wald test you use does fine as well).	We have changed the reporting to "(linear mixed model, factor: location; $\chi^2 = 0.03$, $P = 0.870$; Table S2, $N = 442$ individuals)". We changed all the similar reporting of statistical results in the same manner.
8.	106: not so important for such a low chisquare, still what is df for this test?	The df for the chisquare tests are 1 (two groups compared; $2-1 = 1$). We did not think this was meaningful to report, and we have not added it at this stage (we can add it if deemed necessary). Maybe a brief explanation about why we present chi-square tests at some points will clarify things a bit. The chisquare test is presented when the parameter estimates from the LMM/GLMM models are deemed superfluous or obfuscating. If there is no significant difference, we just present the chisquare test from the ANODEV table (as for body size); when we do not believe strongly in the accuracy of the parameter estimates (as for prevalence; explained in the manuscript, see below) we also only report the chisquare significance. Model summary tables with parameter estimates are, however, always presented in the supplement, so they can be independently evaluated.
9.	107: As a query for body condition you regressed log(mass) on log(total length) as given in 391. This apparently gives a beta difference for the effect of location of -0.022	Using the simple cube law ("Fulton's K"; although Fulton never proposed K; see Nash et al. 2006: The Origin of Fulton's Condition Factor—Setting the Record Straight) would likely work OK for

	(LMM is the test and could be handled as the other test before to keep the annotations similar). You write that body condition is slightly lower? This is however difficult to envision from the beta you give in your model. What exactly means the -0.022? if this means that the relationship between $\log(w)$ and $\log(tl)$ changes than you have a change in the allometric relationship $w = a * tl^b$, where b is commonly around a value of 3. That would be a less steep relationship in the upstream fish? Anyway it gets a bit confusing and I wonder why not use Fulton's condition factor? Then you could show how much that factor has changed from upstream to downstream.	trout, which typically has a b only slightly above 3. However, we prefer to use the more accurate evaluation model, were both a and b are empirically derived for the fish included in the study. Our reported beta-value is the parameter estimate for the difference in a (how much a in upstream fish differ from downstream fish) in the allometric relationship. I.e. it evaluates the intercept a, not the scaling exponent b (our model used the same b for all populations, there is no included term for an interaction between length and location). So it is interpreted as a being 0.22 lower at any given length (i.e. mass is ca. 2.2 % lower in the upstream sections, given that $\exp(-0.022) \approx 0.978$). We have added the estimated percent difference to the text to make it clearer; the rest of the model details as well as the visual illustration of the model are available in the supplement (Table S3 and Fig. S2). The model structure is described in the methods, but it becomes a bit unintuitively presented since the journal format puts the methods at the end. We have noted that we use LMM; LMM is strictly speaking the model, the test of parameter estimates (beta) is a t-test in the lmer-package (the t-value is presented; it is quite rare to see the t-test being explicitly referred to when presenting LMM results, so we have not added this information [similar to not explicitly referring to F-tests when presenting ANOVA results, the presence of a F-value is generally considered enough]).
10.	112: Here you present the data of two analysis in which the second one is on a subset of the data only including rivers in which the Tetracapsuloides was detected. This takes some effort for the reader to understand. Better would be to show only one of the two, and I find the second one more relevant. The other can remain in the supplement. Then first write that xx of xx rivers were found positive for Tetracapsuloides briosalmonae and then do the rest of the analysis only with this subset.	We agree, the first analysis has now been removed from the text, it is provided in the Table S5.
11.	112-113: the use of locations is confusing here. In the model, you use river (equivalent for dam (with a total N of 14)) as random factor. So should this not be N = 13 and N = 9 rivers? In this respect note that you write "Hence, river level variability has a dominant influence on model predictions" in 115!	This probably becomes a bit confusing due to the journal format, where methods are presented at the end of the paper. We have 9 pairs of locations (total N = 18, from N = 9 rivers, in this presented model where we only look rivers with apparent Tb presence [analysing differences in rivers where there is no apparent Tb-infections makes little sense; the difference when there is no Tb at all is

		0 by logic and including such cases obfuscates the results]). The note about river variability refers to the apparent problem for the model to predict good mean effects (it says prevalence is predicted to be above 80 % in upstream locations and 100% for downstream locations, which is far above the median for the pooled data;).
12.	113: Tb-present is a complex annotation. If you first give how many rivers were tested positive you can introduce the annotation. Is that 9 of 14 rivers / dams?	We now added „N = 18 locations (9 pairs/rivers)“.
13.	118: delete "simple" and write that you used mean values per river / location	Done.
14.	119: How do N = 18 locations result in df = 12 in a paired t-test? A df of 12 indicates you had 13 paired values which includes all the rivers (14) -1.	Good point, we have now clarified the use of 13 rivers within the results section as described before.
15.	124: Not clear what the beta would tell me at this point. In case searching for additional information you could consider effect size. Further report which test you used LMM or GLMM and the model p-values etc.	The beta value (parameter estimate) basically represents the effect size; this is the reason why it is included in the text. The beta-value is the estimated average difference between up- and downstream sites, on the sqrt scale. The estimated effect is also presented graphically. LMM would the type of model used (added), the test of the parameter estimate is the t-test (t-value is presented, which should provide the necessary information?) and the full model summary table (including also the intercept p-value from the model) is presented in the referred to Table S6 (and Table S7 for the second model evaluated).
16.	132: Here as for condition you regressed the values. That makes it complex to read the results. Why not test the K/B ratio? Further Kidney hyperplasia is a condition where K/B ratio is high. Consider rewriting this part.	In general it is considered better to not analyse ratios when there is a viable alternative to instead analyse the numerator as dependent on the denominator – that avoids issues of e.g. disregarding how both values contribute to the outcome, the often problematic distributions that ratios have, and the difficulty of understand what a unit of change is at the ratio scale. Ratios are a bit easier to visualise, and many previous papers have analysed the ratios, so we opted to visualise results based on the K/B ratio. Presenting the results in terms of K at a given B should give the equivalent information as the K/B-ratio, with the additional information about how thick the kidney actually is at a given dorsal muscular thickness. The initial sentence of the paragraph is rewritten to clarify that kidney hyperplasia is indicated by a high K/B-ratio.
17.	137: would you still call it kidney hyperplasia here? Better refer to K/B ratio.	Changed accordingly.

18.	137: You mean that the fish in downstream locations of Tb absent rivers have lower K/B ratio than fish in ... Or K/B ratio was lower... etc.	Changed accordingly.
19.	142: What is the definition you use of low parasite loads. Better write descriptive and give the numbers and ranges you found. Similar for 147 Hematocrit.	We have now added further information on the descriptive statistics of parasite load and haematocrit and also adjusted the text, lines 167 to 172.
20.	167: This is the first time you mention a focus on "brown trout recruitment areas" and as it raises confusion about what you mean with this or how it could affect your measurements. This is further only mentioned in the method section. Best would be to add it to the end of the introduction and give some table where we can see how far up and downstream of the dam the data were collected. Moreover, in 232 you correctly mention that waters may naturally warm up in summer while flowing downstream. However, I assume you do not want this to downgrade your central argumentation that it is the damming that increases the water temperature. Now you mention it please argue how this may not explain the extent of increase in temperature to keep your central argumentation intact.	As suggested, we have now added information to the Introduction (lines 98 - 102) regarding the selection of brown trout recruitment areas as close to the dams as possible. We now emphasize that both of these naturally occurring processes may be present, making proximity to the dams crucial for estimation of dam effects on PKD. Since sampling distances from the dams are already presented in Fig. 1B, upstream-downstream averages in text (lines 324 - 325), and distances from the dam for individual sites in Table S1, we did opt not to repeat this information in additional table. We have also added a sentence that described that the measured temperature increase of 2.64 °C over the short river stretch (average distance between sampling points 5.5 km) exceeds what we expect natural processes. Added sentence (Lines 98 to 102): We focused on brown trout recruitment areas located as close to the dams as possible, in order to minimise the influence of natural environmental factors, such as downstream warming or coldwater inputs from groundwater and tributaries, that could confound the assessment of dam and reservoir effects on PKD. Added sentence (Lines 262 - 264): However, the measured water temperature increase (2.64 °C) over the short distances involved (average 5.5 km between sampling points) exceeds what would be expected from natural processes.
21.	188: delete "(class Phylactolaemata)"	Done.
22.	229: There is some new paper on cool water habitat choice in brown trout in relation to PKD, maybe cite that...	Good point, we have added the reference (ref. no. 48, Oexle et al. 2025) to the line 261.
23.	261: The 1C figure does not show: Left: Distribution of dam heights in relation to Tb occurrence. Why are there 16 dams (coloured blocks) in this figure? Also not clear what is the function of the right panel of 1C.	Well spotted. In two rivers there were two consecutive dams, this is described under lines 334 to 336 "In two rivers (R. Mustoja and R. Loobu), two consecutive dams were present (distance between dams 0.65 and 3 km, respectively). Therefore, heights of 16 dams are presented in Fig. 1C and Table S1." We now have described this also in the figure caption. The right

		panel is visualizing the Tb life-cycle and scheme of the effect of temperature on parasite prevalence, load and PKD. We think this visualization is informative to understand the main focus of this study.
24.	285: and figure 1B, is it possible to calculate a relationship between the distance between the two measure points and temperature change with reservoir size as random variable? How does this value stand in relation to the estimated temperature increase due to the reservoir itself?	We explored the reviewer's suggestion by testing whether temperature change between upstream and downstream sites was related to the distance between measurement points, including reservoir size as a covariate. Several linear models were evaluated, with and without an interaction term. None of the models yielded statistically robust effects. The closest to significance was the model including the interaction term, where distance showed a weak trend ($p = 0.058$), but this effect disappeared when the interaction was removed, and no stable patterns emerged across models. In practical terms, the weak trend would correspond to a few degrees difference over very long distances (e.g. ~ 2 °C over 10 km), but given the lack of robustness and sensitivity to model structure, this result is most likely spurious. Overall, we find no consistent or statistically supported relationship between temperature change and distance between sampling sites, nor between temperature change and reservoir size.
25.	400: The square-root transformation should not be between brackets.	The brackets have been removed.
26.	700: In the figure you write renal hyperplasia (also in methods line 328) but in the main text kidney hyperplasia. Keep things consistent...	Well noted, we harmonized the disease trait to renal hyperplasia.
	Reviewer #2 (Remarks to the Author): Dear Authors It was a pleasure to read you MS. It is an interesting and very well-written MS. It tackles a novel and pressing question based on a solid and clever design. I don't have much major comments; all is clear both the data and analyses are strong. My comments are directly embedded into the attached pdf file. Most of these comments can be considered as "minors" but two: (i) the way Tb load is measured (qPCR from kidney DNA) is fine but I would like have a bit more information about whether it is problematic or not to not correct for trout DNA when estimating TB load, and (ii) this response variable might be better modeled using quasipoisson (or negative-binomial) error terms. The the comments in the file for details and more explanation.	We thank the reviewer for the kind words and thorough review. We hereby address the main concern of the reviewer: (i): This is a well-justified consideration, as many studies indeed normalize Tb DNA to trout DNA using qPCR. At the same time, many published works are using total genomic DNA extracted from kidney tissue and measure total genomic DNA spectrophotometrically prior to qPCR quantification of Tb (e.g., Bettge et al., 2009; Hutchins et al., 2018; Lauringson et al. 2025). To reduce variation associated with spectrophotometric DNA quantification, the samples were diluted to $20 \text{ ng } \mu\text{L}^{-1}$, and DNA concentration was subsequently re-measured and adjusted if the estimated values deviated by more than $2 \text{ ng } \mu\text{L}^{-1}$ from the target concentration. Therefore, we are confident that the DNA concentrations were well equalized

	Very nice job, congratulations Best wishes	before qPCR and we added this detail to M&M. Given that the genome size of brown trout (~2.4 Gb) is orders of magnitude larger than genome size of Tb (~18-20 Mb, unpublished genome assembly by Hanna Hartikainen, University of Nottingham) and other typical myxozoan parasites (~20–200 Mb; Chang et al., 2015), and the kidney sample used for DNA isolation consists of mostly trout cells (based on histology studies on Tb), we are confident that total genomic DNA predominantly consists of trout DNA and provides a suitable reference for estimating Tb load via qPCR. Added sentence (Lines 379 - 381): To reduce variation associated with spectrophotometric DNA quantification, DNA dilutions were subsequently re-measured and adjusted if the estimated values deviated by more than 2 ng μL^{-1} from the target concentration. (ii): Answered under point 35.
27.	57: Agreed. Perhaps this paper can be cited as an exception on the impact of weirs on parasite communities: https://pubmed.ncbi.nlm.nih.gov/17647018/	We thank the reviewer for bringing this paper up, we have now added it to the manuscript.
28.	Line 60: Please, indicate explicitly that this is a positive side of dams as this can avoid infection in upstream areas	We have now emphasized that this can be a positive effect on aquatic disease dynamics (line 73 - 74).
29.	63: I would be more integrative as dam restoration does not only include removal. Fishpasses, restoration of an ecologically-suitable flow might also be alternative to removal. I would therefore replace "removal" per "dam ecological restoration" or something like that	Authors response: We have now added "dam removal and restoration of floodplains" (line 76 - 77).
30.	113: If I well understand, there is a huge amount of variability (in prevalence) that is explained by the river identity, and this statistically overestimates the "true" effect of the location (upstream vs. downstream of the dams). Did I interpret it correctly ? If yes (oe even if no) I would advice you to explain this more explicitly, by avoiding precise statistical terms	The reviewers interpretation is not really correct. There is large variability in terms of several rivers aggregating at 100% and 0% prevalence. In proportional data, these values are extreme and here it leads to high (inflated) estimates (expected values) of prevalence (i.e. the 100% values influence the estimation upwards in the modelling procedure) – in contrast, the effect (difference between up- and downstream locations) appears to be underestimated because both up- and downstream estimates appear inflated. We have made an attempt to clarify without too much statistical terminology

		(although it seems impossible to avoid it completely; this is quite technical...)
31.	129: Why is the df different from the t-test presented for the prevalence? This is not logical, it should be the same as the number of replicate is the same	Well noted by the reviewer. It appears that we initially included test results from all rivers (N = 13), rather than restricting the analysis to Tb-positive rivers (N = 9) for the prevalence comparison. We have now corrected the test results on prevalence accordingly.
32.	167: Could you briefly provide some information regarding the distance of the upstream areas to the closest dam upstream? Said in other words; to which extend the selected upstream areas are impacted (or not) by other dams? We expect these upstream areas to be relatively far away from other dams. Having this information would reinforce their "control" status.	Authors response: We thank the reviewer for pointing this out. We added this discussion now to the sub-section of Limitations and Implications of Discussion. In eight of the studied rivers, there are no dams located upstream of the highest sampling location. In six remaining rivers, additional upstream dams were present at different distances from our studied sites. The average distance between the studied upstream location and the nearest upstream dam was 6.1 km. Thus, we cannot rule out that these upstream dams influenced the collected temperature and disease data. Nevertheless, our paired design makes our results robust. Given the well-supported cumulative impacts of multiple reservoirs on water temperature and aquatic ecosystems (e.g. https://doi.org/10.1007/s13201-023-01902-9), we expect that multiple dams may similarly have cumulative effects on disease dynamics. Added text (lines 286 – 292): Due to the high abundance of dams in Estonia, our analyses included rivers with (N = 6) and without (N = 8) additional upstream dams located upstream of our highest sampling sites. The average distance between the studied upstream location and the nearest upstream dam was 6.1 km. Therefore, it is possible that these upstream dams affected the measured variables. However, the paired design ensures the robustness of our results. Given the well-supported cumulative impacts of multiple reservoirs on water temperature and aquatic ecosystems (60), we expect that multiple dams may have similar cumulative effects on disease dynamics.
33.	175: Did you observe some differences in the density of YOY downstream and upstream dams? Could you discuss this a bit?	This is very interesting point but also rather speculative as robust estimation of juvenile salmonid abundance require repeated electrofishing and analyses of sufficient areas and number of locations (eg. https://doi.org/10.1577/03-044). Furthermore, in relation to PKD, it would be perhaps more

		informative to estimate change in juvenile abundance during the first summer but at the same time both mortality and emigration is also influenced by density dependence. We chose not to directly measure YOY density during this study because of time and workload limitations. We also decided not to delve into speculations on mortality in this work although we observed for example a total lack of YOY in two consecutive years below Saesaare dam while still catching older trout specimens. Interestingly, the upstream reservoir of Saesaare dam is the largest among our dataset (48.5 ha). On the other hand, some of the studied downstream sea-trout locations with reasonably high juvenile densities do exhibit severe PKD symptoms. In our previous work (Ahmad et al. 2020), we detected a significant negative correlation of YOY brown 37.trout abundance and summer temperature in R Altja below the dam (Fig 1c, this river was not included to current study because of lack of brown trout above the dam). However, in order to better characterize and separate the effects of PKD and temperature, analyses of long-term electrofishing datasets in combination to annual temperature variation, Tb and PKD data is probably needed.
34.	193: OK, I agree but I think here you need to be more explicit on what should be done next to clearly demonstrate this. For instance, quantifying the abundance of bryozoans above and below dams might be done to confirm your hypothesis. Providing clear guidelines would be more interesting.	We have now emphasized future study directions in relation bryozoans. Added text (lines 218 – 224): Future studies integrating eDNA data with hydrological and thermal profiles could help identify specific habitat features that facilitate bryozoan colonization. Longitudinal studies across seasons and dam types would further clarify how dam-induced habitat changes influence bryozoan abundances, parasite transmission dynamics and disease risk in salmonid populations. These effects may be indirect, via expanded bryozoan habitats that raise parasite spore abundance and infection risk, or direct, through higher disease prevalence and virulence driven by temperature-induced stress in fish.
35.	400: You may consider using GLMM with a quasipoisson (or negative binomial) distribution of error terms, which is often more adequate than a transformation of the raw data as parasite load is generally distributed as count data. You just need to change the decimal numbers to their closest integers	There is indeed a slight issue with the sqrt-transformation, which leads to poor predictability if the model results are applied on comparisons of populations where the load of the parasite is lower – if the parasite load of trout in a river is below ca. 2200 in the downstream section, the load in the upstream section would be predicted

		to be negative on the untransformed scale. Still, we do not think that quasi-Poisson modelling is appropriate; it does not look like our data represent counts that follow a Poisson process, the mean is very high and in such a case a Poisson-distribution should be near-Gaussian, which is not the case (we have a leptokurtic shape). Number of Tb parasites in a fish body might be more determined by an encounter rate function coupled with an infection probability for each parasite, and they may not behave as count data (this is a bit speculative). If we are not mistaken, the quasi-Poisson would scale the SE to the overdispersion, but at high means the assumption about near-Gaussian distribution would still be there (quasi-Poisson only inflate the variance globally, it has no impact on the assumptions of overall distribution shape). Negative binomial is possibly a better option, but applying a proper theta-parameter to describe the skew is tricky without a large amount of data to really find the appropriate underlying data distribution; appropriate theta is central to the neg.bin. modelling. We tested to log-transform the data [$\log_{10}(x+1)$], and then run a linear model on these data. The statistical significance is qualitatively the same ($p \ll 0.00001$) but the mean estimates are lower and confidence intervals are much wider for the downstream group, and confidence intervals for the upstream group are much narrower, as compared to the model on sqrt-transformed data. Overall, the sqrt-model has mean estimates closer to the overall medians and the confidence intervals look more reasonable. Perhaps there are more complex data distributions to apply, but we have here opted for keeping our original model as it seems to represent the overall pattern in terms of differences fairly well.
36.	Reviewer #3 (Remarks to the Author): Comments to authors This study presents a set of very clear results associating dams to infection risk and virulence of Tetracapsuloides bryosalmonae. Association with the PKD disease and temperature is well known but authors are the first to point out that small reservoirs warm up the water and this alone may lead to increase in disease prevalence and virulence. This result is not surprising for those who work in the field but as the data are very convincing this is a very	We thank the reviewer for the thoughtful feedback and for situating our study within a broader ecological and applied context.

	important result for management and restoration of watersheds. It is an especially relevant feedback to the ongoing discussion of the importance of small hydropower plants for sustainable energy production in Europe. Here the message is clear, small hydropower will come with potential disease problems in salmonid fish. It would be very important to have these data in the discussion of watershed management and restoration. I have largely only positive comments on the actual study design, data analysis and interpretation. The design is a solid pairwise upstream-downstream design with sufficient replication. The key result seems large and robust. There is no reason to doubt that these results would not be more general across the PKD / trout range. Methods to detect the parasite, measure the effects of parasite on fish and record the necessary environmental parameters are state of the art and reliable. Statistical analysis is appropriate, but also the effect sizes and the key response seem very clear and robust. Writing is clear and figures are already standalone giving a clear overview of the results. I find this study valuable and well-conducted.	
37.	I have one comment that authors might want to consider. This comment is about direct and indirect effects of temperature increase due to dams. Authors would have the opportunity to deepen the conceptualization of the study by discussing the direct and indirect effects more clearly. I am sure they actually think like that, but it would be good to spell it out. I try to explain what I mean. Readers who know the system will be wondering if the temperature increase benefits the bryozoans that are the intermediate host of the myxozoan parasites. Authors devote a paragraph in the discussion to the possible increase in bryozoan population due to changes in the flow regime and temperature. I think it would be useful to bring the thematic of multiple host life cycle already earlier, even mentioning that in the abstract. One could explain that with multiple host life cycles processes in the intermediate host (indirect temperature effects, for example) may be important. Why do I think this would be important? This would allow authors to explain how direct and indirect effects of	We fully agree that distinguishing between direct and indirect temperature effects, particularly in the context of multiple-host life cycles, offers valuable perspectives and has important implications for management. However, since we did not quantify bryozoan abundances or their fine-scale spatial distribution, we are not in a position to rigorously assess the relative contributions of direct and indirect effects in our study. For this reason, and to keep the focus of the current work aligned with our main research objectives, we have not reframed the introduction or abstract around this theme. That said, following the reviewer's comment, we have expanded the discussion of future research directions to explicitly acknowledge the importance of this distinction. Specifically, we now emphasize the need to better understand dam-induced disease processes through both direct effects (e.g. increased disease prevalence and virulence due to temperature-induced stress in fish) and indirect effects (e.g. enhanced bryozoan habitat suitability and associated increases in parasite spore abundance and

increasing temperature operate in this system. PKD is an example of a parasite which has higher virulence in stressed host (trout), and here authors measure virulence. Temperature will increase the stress levels of fish. PKD is also an example of a parasite where parasite risk depends on density of infected bryozoans in the environment. Bryozoans are filter feeders that thrive in rivers with pools of slowly flowing water, solid (built) structures and low water level variation. Bryozoans probably like impoundments like authors write. If there is something missing in the study it is the assessment of the density of infected and uninfected bryozoans upstream, in reservoirs and downstream. It would be a great addition to the study, but probably authors did not sample bryozoans. I might be asking a bit too much, but it would be great to be able to dissect / discuss the direct and indirect effects of the temperature. Direct effects would be the increase in disease prevalence and virulence due to stressed fish (temperature dependent susceptibility and virulence). Indirect effects would be the epidemiological consequences of more favorable bryozoan habitats (increase in parasite spore numbers and infection risk). This distinction might have importance for management decisions. Authors could also write the introduction and motivation of the study by explaining the expectations with respect to direct and indirect effects of temperature, showing that the problematic with abiotic stressors comes with indirect effects through ecological change in the environment.	infection risk). We believe this addition provides a forward-looking perspective while maintaining the primary focus of the present study. Added text (lines 218 – 224): Future studies integrating eDNA data with hydrological and thermal profiles could help identify specific habitat features that facilitate bryozoan colonization. Longitudinal studies across seasons and dam types would further clarify how dam-induced habitat changes influence bryozoan abundances, parasite transmission dynamics and disease risk in salmonid populations. These effects may be indirect, via expanded bryozoan habitats that raise parasite spore abundance and infection risk, or direct, through higher disease prevalence and virulence driven by temperature-induced stress in fish.
---	---

1 **Title:** Dams threaten salmonids by triggering temperature-dependent disease

Magnus Lauringson¹, Joacim Näslund², Lilian Pukk¹, Siim Kahar¹, Riho Gross¹, Anti
Vasemägi^{1,2*}

¹Chair of Aquaculture, Institute of Veterinary Medicine and Animal Sciences, Estonian
University of Life Sciences, 46A Kreutzwaldi St., 51006 Tartu, Estonia

²Department of Aquatic Resources, Institute of Freshwater Research, Swedish University of
Agricultural Sciences, Stångholmsvägen 2, 17893 Drottningholm, Sweden

Correspondence: Anti Vasemägi

**Email:** anti.vasemagi@slu.se

**Competing Interest Statement:** Authors declare that they have no competing interests.

**Classification:** Biological Sciences, Environmental Sciences

**Keywords:** Climate change, fish disease, emerging disease, parasites, habitat degradation

Abstract

Dams are ubiquitous in river systems, providing essential services such as drinking water,
electricity and irrigation to human society. At the same time, dams significantly impact ecosystems
by disrupting flow, and altering natural water temperature regimes. Here, we describe a novel,
unappreciated threat posed by dams and reservoirs to one of the world's most popular game fish,
brown trout (*Salmo trutta*). We show that small river impoundments elevate downstream water
temperature in summer, which increase the prevalence and abundance of malacosporean parasite
*Tetracapsuloides bryosalmonae* triggering proliferative kidney disease (PKD), an emerging
disorder in salmonids across North America and Europe. Our study highlights the role of reservoirs
in creating parasite and disease hotspots, while providing limited evidence that dams act as barriers
to *T. bryosalmonae* spread. As global temperatures continue to rise, reservoirs are likely to cause
unsustainable effects on downstream riverine fishes through temperature-induced diseases, with
particularly severe consequences for cold-water salmonids. This makes downstream areas from
reservoirs valuable sentinel sites for monitoring climate impacts on riverine ecosystems.
Ultimately, the assessment of dams requires a more holistic approach, where the disease risks are
included in the decision-making process balancing human needs with the health of aquatic
ecosystems.

Main Text

Introduction

[revised manuscript text omitted]

90 91 **Results**

93 ***Water temperature and body size***

Compared to upstream river sections, the water temperature was consistently higher downstream
of the reservoirs in all studied cases ($N = 14$ dams/reservoirs, Fig. 1, A and D, Table S1). On
average, water temperature was 2.64 °C higher (95% CI: 1.69 to 3.58 °C, range: 0.24 to 5.73 °C)
downstream of the dams, as compared to the upstream location (Wilcoxon signed-rank test $P <$
0.001; Fig. 1D). Similarly, downstream areas exhibited water temperature over 15 °C (clinical PKD
threshold) for longer time period compared to upstream locations (average no. of days >15 °C, up-
vs. downstream: 33.21 vs 51.3, Wilcoxon signed-rank test $P < 0.001$; Fig 1E, Table S1). There was
no difference in diurnal water temperature fluctuation between upstream and downstream locations
(up- vs. downstream: 2.05 °C vs. 1.96 °C, Wilcoxon signed-rank test $P = 0.552$, Fig. S1, Table S1).
The extent of downstream warming was positively associated with reservoir size, though this did
not reach statistical significance (β : 1.3576, 95% CI: -0.514 to -2.592, $P = 0.074$, LM; Fig. 1F).
Body size did not differ significantly between locations upstream and downstream of dams ($N = 13$
dams), neither in total length (ANODEV (location): $\chi^2 = 0.03$, $P = 0.870$; Table S2, $N = 442$

individuals). Body mass at a given length (body condition), however, was on average slightly lower
in the upstream locations ($\beta_{\text{upstream}}: -0.022 \pm 0.01 \text{ SE}, t = -2.83, P = 0.005$; Table S3, Fig. S2).

***Infection prevalence and parasite load***

Infection prevalence was significantly lower upstream of dams compared to downstream locations
(ANODEV_{all data (location)}: $\chi^2 = 42.3, P < 0.001, N = 26$ locations, Table S4; ANODEV_{Tb-present rivers}
(location): $\chi^2 = 42.1, P < 0.001, N = 18$ locations, Table S5). A closer look at the GLMM for *Tb*-
present rivers show that $SD = 5.72$ for the random effect, compared to the fixed effect ($\beta_{\text{upstream}} = -$
4.97), indicating substantial variability in baseline log-odds of the response variable. Hence, river-
level variability has a dominant influence on model predictions, where expected values become
inflated (Fig. 2A) such that the model overestimates the mean tendency when compensating for
high variability. Using a simple pairwise *t*-test to compare infection prevalence upstream and
downstream locations in *Tb* present rivers, the differences are still significant ($t = 2.40, df = 12, P$
$= 0.034, N = 18$ locations).

Parasite load, quantified as copies of *Tb* fragments per qPCR reaction was generally higher
downstream of the dams (Fig. 2B), with effects detected using both the full data set ($\beta_{\text{upstream}}: -47.2$
$\pm 5.1 \text{ SE}, t = -9.30, P < 0.001, N = 444$ individuals, Table S6) and the reduced set excluding
apparently *Tb*-absent rivers ($\beta_{\text{upstream}}: -65.4 \pm 6.8 \text{ SE}, t = -9.60, P < 0.001, N = 315$ individuals,
Table S7). The maximum parasite loads for each river are presented in Fig. S3 and the average
values (\pm standard error) for each river are presented in Fig. S4. Using a simple pairwise *t*-test to
compare parasite load upstream and downstream locations in *Tb* present rivers, the differences are
still significant ($t = 2.1, df = 8, P = 0.034, N = 18$ locations).

***Kidney hyperplasia and anemia***

Kidney hyperplasia, measured as K/B ratio and analysed as K at a given B, was more severe in the
downstream locations ($\beta_{\text{downstream}}: 0.50 \pm 0.04 \text{ SE}, t = 11.8, P < 0.001, N = 443$ individuals, Table
S8). From the follow-up model (Table S9), the latter result was verified in the *Tb* present rivers
[estimated difference at average body length (upstream – downstream): $-0.64 \text{ mm}, t = -13.1, P <$
0.001], but not in *Tb* absent rivers (estimated difference: $-0.14 \text{ mm}, t = -1.70, P = 0.323$).
Downstream locations in *Tb* absent rivers had lower levels of kidney hyperplasia than downstream
locations in *Tb* present rivers (estimated difference: $-0.78 \text{ mm}, t = -3.22, P = 0.032$) and the same
was the case in the upstream locations (estimated difference: $-0.92 \text{ mm}, t = -3.82, P = 0.012$) (Fig.
2, C and D, Fig. S4).

The K/B-ratio started to increase already at low parasite loads and the average K/B-ratio plateaued
at parasite loads above approximately 8,000 *Tb* copies/reaction ($N = 213$ individuals, Fig. 3A).
After plateauing, there was a substantial variation in K/B-ratio, but no values were found to be
lower than the average for fish in *Tb* absent rivers.

Hematocrit was found to be generally lower at the highest parasite loads ($N = 209$ individuals, Fig.
3B). Hematocrit showed a clear negative relationship with K/B-ratio in *Tb* present rivers, which
clearly deviates from the uninfected individuals ($N = 217$ individuals, Fig. 3C). Severe anemia
(hematocrit: < 0.25) was observed among individuals with enlarged kidneys ($N = 27$ individuals,
K/B-ratio: > 0.2).

[revised manuscript text omitted]

Infection prevalence (proportion of infected individuals; pv) was analysed with the GLMM: $\text{pv} \sim$
$\text{location} + (1|\text{river})$, weights = tot.obs, family = binomial. For weights, we used the total number of
individuals (tot.obs) that each location-specific proportion value was based on. Overdispersion was
tested through the *DHARMA* package (76) and found non-significant (dispersion = 1.14, $p = 0.30$).
The same model was run using a data subset including only rivers where the parasite was detected,
given that effects are mainly of interest in rivers with confirmed parasite presence.

Parasite load (pl) was analysed (square-root transformed to accommodate for positive skew) with
the LMM: $\text{sqrt(pl)} \sim \text{location} + (1|\text{river})$; focusing on comparing upstream and downstream sites
overall. We also ran the same model using a data subset including only rivers where the parasite
was detected.

Renal hyperplasia was analysed as the relative thickness of the kidney to dorsal muscular tissue
(Fig. 1J). In models, *kidney thickness* (k) was the dependent variable, height of dorsal musculature
(*body thickness*; b) was added as a covariate [LMM: $k \sim b + \text{location} + (1|\text{river})$] representing a
proxy of the overall size of the fish; in data plots, we use the calculated K/B-ratio ('K/B-ratio' =
k/b). A follow-up model was built utilizing the fact that some rivers ($N = 4$) were completely
parasite-free in the analysed samples (*ad hoc* controls), adding the factor *parasite presence*
(par.pres) in interaction with location [model: $k \sim b + \text{location} \times \text{par.pres} + (1|\text{river})$]. Results from
the latter model were analysed using pairwise location \times par.pres contrasts (Kenward-Roger method
for degrees-of-freedom; Tukey adjustment for multiple comparisons), using the *emmeans* package
(77).

Progression of renal hyperplasia (as K/B-ratio) in relation to parasite load was analysed for infected
individuals, using a non-linear loess regression (pooling all rivers). The relationship between the
variables was assessed graphically based on the 95% confidence band relative to the estimated mean
K/B-ratio for uninfected individuals (non-overlapping confidence bands indicating significant
difference).

Hematocrit was analysed in relation to parasite load and renal hyperplasia using non-linear loess
regressions, as above. Reservoir area and downstream temperature rise was assessed using linear
regression.

426 427 **Acknowledgments**

We thank Alfonso Diaz Suarez, Gustav Lauringson, Karl-Erik Aavik, Tanel Ader and Veljo
Kisand for their assistance during fieldwork, Põlula Fish Rearing Centre of State Forest
Management Centre (RMK) for logistical support and Oksana Burimski for laboratory assistance.
Vihula III dam and reservoir photo (10.6 ha) at the Mustoja river, shown in Fig. 1G, is based on a
drone photo from 10.07.2023, kindly provided by Gunnar Laak. The study was funded by
Estonian Ministry of Regional Affairs and Agriculture, contract no. 4-1/22/6, Estonian Research
Council, grant no. PSG849, PRG852 Swedish research council for sustainable development
FORMAS, grant no. 2021-01643.

1. Belletti, B., De Leaniz, C. G., Jones, J., Bizzi, S., Börger, L., Segura, G., Castelletti, A., Van De Bund, W., Aarestrup, K., Barry, J., Belka, K., Berkhuisen, A., Birnie-Gauvin, K., Bussettini, M., Carolli, M., Consuegra, S., Dopico, E., Feierfeil, T., Fernández, S., Zalewski, M. More than one million barriers fragment Europe's rivers. *Nature* **588**, 436–441 (2020).
2. Grill, G., Lehner, B., Thieme, M., Geenen, B., Tickner, D., Antonelli, F., Babu, S., Borrelli, P., Cheng, L., Crochetiere, H., Macedo, H. E., Filgueiras, R., Goichot, M., Higgins, J., Hogan, Z., Lip, B., McClain, M. E., Meng, J., Mulligan, M., Zarfl, C. Mapping the world's free-flowing rivers. *Nature* **569**, 215–221 (2019).
3. Zarfl, C., Berlekamp, J., He, F., Jähnig, S. C., Darwall, W., Tockner, K. Future large hydropower dams impact global freshwater megafauna. *Sci. Rep.* **9** (2019).
4. Su, G., Logez, M., Xu, J., Tao, S., Villéger, S., Brosse, S. Human impacts on global freshwater fish biodiversity. *Science* **371**, 835–838 (2021).
5. Auffray, M., Senécal, J.-F., Turgeon, K., St-Hilaire, A., Maheu, A. Reservoirs regulated by small dams have a similar warming effect than lakes on the summer thermal regime of streams. *Sci. Total Environ.* **869**, 161445 (2023).
6. Bednarek, A. Undamming rivers: a review of the ecological impacts of dam removal. *Environ. Manage.* **27**, 803–814 (2001).
7. Barbarossa, V., Schmitt, R. J. P., Huijbregts, M. A. J., Zarfl, C., King, H., Schipper, A. M. Impacts of current and future large dams on the geographic range connectivity of freshwater fish worldwide. *Proc. Natl. Acad. Sci. U.S.A.* **117**, 3648–3655 (2020).
8. Palmer, M. A., Reidy Liermann, C. A., Nilsson, C., Flörke, M., Alcamo, J., Lake, P. S., Bond, N. Climate change and the world's river basins: anticipating management options. *Front. Ecol. Environ.* **6**, 81–89 (2008).
9. Deemer, B. R., Harrison, J. A., Li, S., Beaulieu, J. J., DelSontro, T., Barros, N., Bezerra-Neto, J. F., Powers, S. M., Santos, M. A. D., Vonk, J. A. Greenhouse Gas Emissions from Reservoir Water Surfaces: A New Global Synthesis. *BioScience* **66**, 949–964 (2016).
10. Olden, J. D., Naiman, R. J. Incorporating thermal regimes into environmental flows assessments: modifying dam operations to restore freshwater ecosystem integrity. *Freshwater Biol.* **55**, 86–107 (2010).
11. Sinokrot, B. A., Stefan, H. G., McCormick, J. H., Eaton, J. G. Modeling of climate change effects on stream temperatures and fish habitats below dams and near groundwater inputs. *Climatic Change* **30**, 181–200 (1995).
12. Zaidel, P. A., Roy, A. H., Houle, K. M., Lambert, B., Letcher, B. H., Nislow, K. H., Smith, C. Impacts of small dams on stream temperature. *Ecol. Indic.* **120**, 106878 (2021).
13. Pereira, H. R., Gomes, L. F., Barbosa, H. D., Agostinho, A. A. Research on dams and fishes: Determinants, directions, and gaps in the world scientific production. *Hydrobiologia* **847**, 579–592 (2020).
14. Marcos-López, M., Gale, P., Oidtmann, B. C., Peeler, E. J. Assessing the impact of climate change on disease emergence in freshwater fish in the United Kingdom. *Transbound. Emerg. Dis.* **57**, 293–304 (2010).
15. Bartholomew, J. L., Alexander, J. D., Alvarez, J., Atkinson, S. D., Belchik, M., Bjork, S. J., Foott, J. S., Gonyaw, A., Hereford, M. E., Holt, R. A., McCovey, B., Som, N. A., Soto, T., Voss, A., Williams, T. H., Wise, T. G., Hallett, S. L. Deconstructing dams and disease: predictions for salmon disease risk following Klamath River dam removals. *Front. Ecol. Evol.* **11** (2023).
16. Ros, A., Schmidt-Posthaus, H., Brinker, A. Mitigating human impacts including climate change on proliferative kidney disease in salmonids of running waters. *J. Fish Dis.* **45**, 497–521 (2022).

[revised manuscript text omitted]

34. Carraro, L., Bertuzzo, E., Mari, L., Fontes, I., Hartikainen, H., Strepparava, N., Schmidt-Posthaus, H., Wahli, T., Jokela, J., Gatto, M., Rinaldo, A. Integrated field, laboratory, and theoretical study of PKD spread in a Swiss prealpine river. *Proc. Natl. Acad. Sci. U.S.A.* 114, 11992–11997 (2017).
35. Ahmad, F., Debes, P. V., Nousiainen, I., Kahar, S., Pukk, L., Gross, R., Ozerov, M., Vasemägi, A. The strength and form of natural selection on transcript abundance in the wild. *Mol. Ecol.* 30, 2724–2737 (2020).
36. Bailey, C., Segner, H., Casanova-Nakayama, A., Wahli, T. Who needs the hotspot? The effect of temperature on the fish host immune response to *Tetracapsuloides bryosalmonae* the causative agent of proliferative kidney disease. *Fish Shellfish Immunol.* 63, 424–437 (2017).
37. Bruneaux, M., Visse, M., Gross, R., Pukk, L., Saks, L., Vasemägi, A. Parasite infection and decreased thermal tolerance: impact of proliferative kidney disease on a wild salmonid fish in the context of climate change. *Funct. Ecol.* 31, 216–226 (2016).
38. Nagrodski, A., Raby, G. D., Hasler, C. T., Taylor, M. K., Cooke, S. J. Fish stranding in freshwater systems: Sources, consequences, and mitigation. *J. Environ. Manage.* 103, 133–141 (2012).
39. Huntingford, F. A., Aird, D., Joiner, P., Thorpe, K. E., Braithwaite, V. A., Armstrong, J. D. How juvenile Atlantic salmon, *Salmo salar* L., respond to falling water levels: experiments in an artificial stream. *Fish. Manage. Ecol.* 6, 357–364 (1999).
40. Jonsson, B., Jonsson, N. A review of the likely effects of climate change on anadromous Atlantic salmon *Salmo salar* and brown trout *Salmo trutta*, with particular reference to water temperature and flow. *J. Fish Biol.* 75, 2381–2447 (2009).
41. Schisler, G. J., Walker, P. G., Chittum, L. A., Bergersen, E. P. Gill ectoparasites of juvenile rainbow trout and brown trout in the upper Colorado River. *J. Aquat. Anim. Health* 11, 170–174 (1999).
42. Marcogliese, D. J. The impact of climate change on the parasites and infectious diseases of aquatic animals. *Rev. Sci. Tech.* 27, 467–484 (2008).
43. Anderson, R. M., May, R. M. The invasion, persistence and spread of infectious diseases within animal and plant communities. *Philos. Trans. R. Soc. Lond. B Biol. Sci.* 314, 533–570 (1986).
44. Schmidt-Posthaus, H., Schneider, E., Schölzel, N., Hirschi, R., Stelzer, M., Peter, A. The role of migration barriers for dispersion of Proliferative Kidney Disease—Balance between disease emergence and habitat connectivity. *PLoS ONE* 16, e0247482 (2021).
45. Bellmore, J. R., Pess, G. R., Duda, J. J., O'Connor, J. E., East, A. E., Foley, M. M., Wilcox, A. C., Major, J. J., Shafroth, P. B., Morley, S. A., Magirl, C. S., Anderson, C. W., Evans, J. E., Torgersen, C. E., Craig, L. S. Conceptualizing ecological responses to dam removal: If you remove it, what's to come? *BioScience* 69, 26–39 (2018).
46. Dolan, E. J., Soto, I., Dick, J. T. A., He, F., Cuthbert, R. N. Riverine barrier removals could proliferate biological invasions. *Glob. Change Biol.* 31 (2025).
47. Altizer, S., Bartel, R., Han, B. A. Animal migration and infectious disease risk. *Science* 331, 296–302 (2011).
48. Kalny, G., Laaha, G., Melcher, A., Trimmel, H., Weihs, P., Rauch, H. P. The influence of riparian vegetation shading on water temperature during low flow conditions in a medium sized river. *Knowl. Manage. Aquat. Ecosyst.* 418, 5 (2017).
49. Wohl, E., Lane, S. N., Wilcox, A. C. The science and practice of river restoration. *Water Resour. Res.* 51, 5974–5997 (2015).

- 50. Myrstener, M., Greiser, C., Kuglerová, L.. Downstream temperature effects of boreal forest
clearcutting vary with riparian buffer width. *Water Resour. Res.*, **61**, e2024WR037705
(2025).
- 51. Wahli, T., Bernet, D., Steiner, P. A., Schmidt-Posthaus, H. Geographic distribution of
*Tetracapsuloides bryosalmonae* infected fish in Swiss rivers: an update. *Aquat. Sci.* **69**, 3–
10 (2007).
- 52. Mo, T. A., Jørgensen, A. A survey of the distribution of the PKD-parasite *Tetracapsuloides*
*bryosalmonae* (Cnidaria: Myxozoa: Malacosporea) in salmonids in Norwegian rivers –
additional information gleaned from formerly collected fish. *J. Fish Dis.* **40**, 621–627
(2016).
- 53. Gorgoglione, B., Bailey, C., Ferguson, J. A. Proliferative kidney disease in Alaskan
salmonids with evidence that pathogenic myxozoans may be emerging north. *Int. J.*
*Parasitol.* **50**, 797–807 (2020).
- 54. Philpott, D., Näslund, J., Donadi, S., Burimski, O., Lauringson, M., Pukk, L., Vasemägi, A.
Effects of different preservatives during ecological monitoring of malacosporean parasite
*Tetracapsuloides bryosalmonae* causing proliferative kidney disease (PKD) in salmonids.
*J. Fish Dis.* e14095 (2025).
- 55. Hutchins, P. R., Sepulveda, A. J., Hartikainen, H., Staigmiller, K. D., Opitz, S. T.,
Yamamoto, R. M., Huttinger, A., Cordes, R. J., Weiss, T., Hopper, L. R., Purcell, M. K.,
Okamura, B. Exploration of the 2016 Yellowstone River fish kill and proliferative kidney
disease in wild fish populations. *Ecosphere* **12** (2021).
- 56. Schager, E., Peter, A., Burkhardt-Holm, P. Status of young-of-the-year brown trout (*Salmo*
*trutta fario*) in Swiss streams: factors influencing YOY trout recruitment. *Aquat. Sci.* **69**,
41–50 (2007).
- 57. Waldner, K., Bechter, T., Auer, S., Borgwardt, F., El-Matbouli, M., Unfer, G. A brown trout
(*Salmo trutta*) population faces devastating consequences due to proliferative kidney
disease and temperature increase: A case study from Austria. *Ecol. Freshw. Fish* **29**, 465–
476 (2019).
- 58. Szklo, M., Nieto, F. J. Epidemiology: Beyond the basics (3rd ed.). Jones & Bartlett
Learning, p. 139 (2014).
- 59. Carolli, M., De Leaniz, C. G., Jones, J., Belletti, B., Hušek, H., Pusch, M., Pandakov, P.,
Börger, L., Van De Bund, W. Impacts of existing and planned hydropower dams on river
fragmentation in the Balkan Region. *Sci. Total Environ.* **871**, 161940 (2023).
- 60. Fiske, P., Forseth, T., Thorstad, E. B., Bakkestuen, V., Einum, S., Falkegård, M., Garmo,
Ø. A., Garseth, Å. H., Skoglund, H., Solberg, M., Utne, K. R., Vollset, K. W., Vøllestad, L.
621 A., Wennevik, V. Novel large-scale mapping highlights poor state of sea trout populations.
*Aquat. Conserv. Mar. Freshw. Ecosyst.* **34** (2024).
- 61. Löhmus, M., Björklund, M. Climate change: what will it do to fish-parasite interactions?
*Biol. J. Linn. Soc.* **116**, 397–411 (2015).
- 62. Lauringson, M., Kahar, S., Veevo, T., Silm, M., Philpott, D., Svirgsden, R., Rohtla, M.,
Pääk, P., et al. Spatial and intra-host distribution of myxozoan parasite *Tetracapsuloides*
*bryosalmonae* among Baltic sea trout (*Salmo trutta*). *J. Fish Dis.* **46**, 1073–1083 (2023).
- 63. Dash, M., Vasemägi, A. Proliferative kidney disease (PKD) agent *Tetracapsuloides*
*bryosalmonae* in brown trout populations in Estonia. *Dis. Aquat. Org.* **109**, 139–148 (2014).
- 64. Sergeant, E. S. G. Epitools Epidemiological Calculators. Ausvet. Available at:
<http://epitools.ausvet.com.au> (2018).
- 65. Debes, P., Gross, R., Vasemägi, A. Quantitative genetic variation in, and environmental
effects on, pathogen resistance and temperature-dependent disease severity in a wild trout.
*Am. Nat.* **190**, 000–000 (2017).

[revised manuscript text omitted]

**Supplementary Materials**

**Figs. S1 to S4**

Supplementary Fig. 1. Diurnal water temperature changes up- vs. downstream of the dams.

Supplementary Fig. 2. Residual body mass for downstream and upstream locations.

Supplementary Fig. 3. Maximum parasite load (*Tb* copies/reaction) up- and downstream of dams.

Supplementary Fig. 4. Disease traits, parasite load and prevalence of individual rivers.

**Tables S1 to S10**

Supplementary Table 1. Sampling site information with dam/reservoir characteristics, parasite
prevalence and temperature estimates.

Supplementary Table 2. Summary table from linear mixed model for analysis of total length.

Supplementary Table 3. Summary table from linear mixed model for analysis of body condition.

Supplementary Table 4. Summary table from binomial generalized mixed model (logit-link) for
analysis of parasite prevalence (all rivers).

Supplementary Table 5. Summary table from binomial generalized mixed model (for analysis of
parasite prevalence (only including *Tb* present rivers).

Supplementary Table 6. Summary table from linear mixed model for analysis of parasite load (all
rivers).

Supplementary Table 7. Summary table from linear mixed model for analysis of parasite load (only
including *Tb* present rivers).

Supplementary Table 8. Summary table from linear mixed model for analysis of renal hyperplasia
(all data).

Supplementary Table 9. Summary table from linear mixed model for analysis of renal hyperplasia.
Model includes factor parasite presence in interaction with location to compare *Tb* present/absent
rivers.

Supplementary Table 10. Estimated qPCR parameters associated with *Tb* quantification.

Fig. S1: Diurnal water temperature changes up- vs. downstream of the dams.

Fig. S2: Residual body mass for downstream and upstream locations. Residuals from model: $tm \sim \log(tl) + \text{location} + (1|\text{river})$. Red horizontal line shows the overall mean.

Fig. S3: Maximum parasite load (*Tb* copies/reaction) up- and downstream of dams.

 **Fig. S4:** Summarizing figure of renal hyperplasia (K/B-ratio; kidney thickness divided by dorsal
 muscle thickness), parasite loads (Load; *Tb* copies/reaction) and prevalence (% infected
 individuals) in brown trout in Estonian rivers (upstream and downstream of dams). Parasite loads
 are summarised for each river, in terms of mean and standard deviation ($\mu \pm SD$), minimum and
 maximum values (min/max), and number of individuals with detected loads out of total sample
 ('Preval.'). For each river location, the prevalence is also indicated by graph background shading.
 The K/B-ratio is presented for each river (points: individual fish; horizontal bar = mean), as well as
 in the form of estimated marginal means with 95% confidence intervals (lower right); in the latter
 graph, the estimates relate to the mean kidney thickness (mm) at the mean body thickness.

**Table S1:** Sampling site information with dam/reservoir characteristics, parasite prevalence and temperature estimates.

River - Dam name	Date (dd.mm.yyyy)	Location from dam	Coordinates (N, E)	Distance from the dam (km)	No. of juveniles sampled	No. of infected	No. of uninfected	Tb prev. (95% CI)	Parasite load (Tb copies/reaction)	K/B ratio	Hematocrit	Dam height (m)	Reservoir size (ha)	Average water temperature (°C)	No. Of days > 15 °C	Diurnal water temp. difference (°C)
Rannametsa - Laiksaare	22.08.2022	Downstream	58°05'46.5, 24°40'02.7"	0.1	20	0	20	0 (0–0.161)	0	0.127	0.329	3.35	1.2	19.68	55	2.75
Rannametsa – Laiksaare	22.08.2022	Upstream	58°05'40.3, 24°40'23.6"	0.4	20	0	20	0 (0–0.161)	0	0.105	0.336			16.45	45	1.48
Vigala – Kuusiku	23.08.2022	Downstream	58°57'54.3, 24°43'01.4"	0.1	10	0	10	0 (0–0.277)	0	0.098	0.368	2.15	5.4	17.51	55	0.77
Vigala – Kuusiku	23.08.2022	Upstream	58°55'25.7, 24°51'09.0"	11.8	15	0	15	0 (0–0.204)	0	0.082	0.299			12.20	2	2.81
Vainupea – Pajuveski	24.08.2022	Downstream	59°34'02.4, 26°15'28.7"	0.2	20	6	14	0.2 (0.145–0.52)	1404.7	0.139	0.321	2.65	0.6	19.47	55	2.33
Vainupea – Pajuveski	24.08.2022	Upstream	59°34'02.4, 26°15'28.7"	1.8	20	0	20	0 (0–0.161)	0	0.110	0.341			17.50	53	2.14

Selja - Päide	25.08.20 22	Downst ream	59°23' 27.4", 26°23' 30.4" 59°22' 31.5,	7.7	20	15	5	0.75 (0.531– 0.888)	699.5	0.1 18	0.379	2.2 5	3.1	18.2 9	55	2.29
Selja - Päide	25.08.20 22	Upstrea m	26°17' 13.6" 59°33' 09.9,	2.9	20	0	20	0 (0–0.20 4)	0	0.1 04	0.333			14.1 4	14	2.47
Mustoja - Vihula II/III	25.08.20 22	Downst ream	26°10' 59.3" 59°31' 48.0,	1.5	20	100	0	1 (0.839– 1)	44255	0.3 31	0.29	2.5 5/6	2.2/1 0.6	15.7 2	55	2.63
Mustoja - Vihula II/III	26.08.20 22	Upstrea m	26°10' 44.4" 59°00' 22.7,	2	20	1	19	0.05 (0–0.23 6)	5.4	0.0 96	0.382			13.3 8	48	2.34
Põltsam aa – Ao	27.08.20 22	Downst ream	26°12' 27.7" 59°03' 30.8,	0.1	6	0	6	0 (0–0.39)	0	0.1 18	0.318	0.9 5	10.3	19.2 4	55	1.41
Põltsam aa – Ao Nõmme	27.08.20 22	Upstrea m	26°10' 38.8" 59°02' 32.8,	9.9	20	0	20	0 (0–0.16 1)	0	0.0 88	0.359			13.5 0	4	1.61
Nõmme veski Nõmme	28.08.20 22	Downst ream	26°13' 29.7" 59°02' 42.1,	0.2	18	0	18	0 (0–0.17 6)	0	0.1 07	0.35	3.2	5.7	15.7 2	34	2.17
Nõmme veski	28.08.20 22	Upstrea m	26°14' 20.1"	1	20	0	20	0 (0–0.16 1)	0	0.1 06	0.313			13.3 8	7	1.55
Ahja – Saesaare	29.08.20 22	Downst ream	58°06' 52.2,	0.6	0	N/A	N/A	N/A	N/A	N/ A	N/A	7.6 5	48.5	19.0 2	55	1.15

Ahja – Saesaare	29.08.20 22	Upstrea m	27°03' 05.5" 58°08' 38.9, 26°58' 28.9"	12.6	15	15	0	1 (0.796– 1)	10938.1	0.1 21	0.34			16.5 7	50	1.03
Võhand u – Hutita	31.08.20 22	Downst ream	00.1, 26°44' 32.5" 57°53'	0.1	19	19	0	1 (0.832– 1)	35690	0.2 42	0.339	2.2 5	2.1	16.8 7	50	1.6
Võhand u – Hutita	29.08.20 22	Upstrea m	32.7, 26°45' 05.5" 57°45'	7.4	20	20	0	1 (0.839– 1)	44827.5	0.1 98	0.297			15.9 2	42	1.94
Pärlijõgi - Alaveski	31.08.20 22	Downst ream	23.6, 26°45' 55.6" 59°02'	4.3	20	20	0	1 (0.839– 1)	35368.9	0.2 34	0.329	1	0.5	15.6 3	36	2.16
Pärlijõgi - Alaveski	31.08.20 22	Upstrea m	42.1, 26°14' 20.1" 58°08'	1.1	10	10	0	1 (0.722– 1)	10468	0.1 41	0.407			15.3 9	32	2.57
Elva - Hellenur me	01.09.20 22	Downst ream	33.8, 26°23' 51.7" 58°06'	1	21	21	0	1 (0.845– 1)	19150.3	0.1 83	0.384	2.9	5.5	18.9 9	55	2.7
Elva - Hellenur me	01.09.20 22	Upstrea m	54.8, 26°23' 17.7" 57°45'	3.8	17	17	0	1 (0.816– 1)	897.9	0.1 09	0.368			16.1 0	41	2.52
Pedja – Käruves ki	02.09.20 22	Downst ream	23.6, 26°45' 55.6"	3.6	12	12	0	1 (0.757– 1)	80640	0.1 9	0.391	1	1	17.4 5	53	1.85

Pedja – Käruveski Vasalemma – Töökmani Vasalemma – Töökmani	02.09.20 22	Upstream	58°56' 52.8, 26°30' 25.4"	5.7	20	20	0	1 (0.839– 1)	19100.2	0.1 87	0.36			15.3 5	32	1.99
	05.09.20 22	Downstream	59°12' 25.1, 24°25' 06.7"	0.3	10	6	4	0.6 (0.313– 0.832)	2224.1	0.1 12	0.443	1	0.6	17.1 9	51	1.8
	05.09.20 22	Upstream	59°12' 10.4, 24°26' 10.6"	1.1	10	1	9	0.1 (0–0.04)	21	0.1 03	0.346			16.9 7	48	2.03
Loobu - Undla/K adrina	30.08.20 23	Downstream	59°21' 00.5, 26°06' 31.8"	0.2	23	23	0	1 (0.857– 1)	5903.8	0.2 97	0.295	1/2. 65	0.5/1 5.5	18.7 0	55	1.89
Loobu - Undla/K adrina	30.08.20 23	Upstream	59°18' 46.5, 26°09' 53.4"	2.5	15	2	13	0.13 (0–0.03 8)	1.3	0.0 92	0.377			16.3 5	47	2.19

1 **Table S2:** Summary table from linear mixed model for analysis of total length (mm): tl
 2 ~location + (1|river).

Random effects

Groups		Variance	SD
River	(Intercept)	85.9	9.3
Residual		106.9	10.3

N (obs.): 442

N (River): 13

Fixed effects

	Estimate	SE	df	t	p
(Intercept)	74.2	2.7	13.1	27.8	5.17e-13
Location (upstream)	-0.16	1.00	429.4	-0.16	0.870

3

4 **Table S3:** Summary table from linear mixed model for analysis of body condition (relative
 5 total mass, g): $tm \sim \log(tl) + \text{location} + (1|\text{river})$.

Random effects

Groups		Variance	SD
River	(Intercept)	0.002	0.042
Residual		0.007	0.081

N (obs.): 442

N (River): 13

Fixed effects

	Estimate	SE	df	t	p
(Intercept)	-12.32	0.12	371.2	-106.7	< 2e-16
$\log_e(\text{total length})$	3.18	0.03	388.8	118.9	< 2e-16
Location (upstream)	-0.02	0.01	430.0	-2.83	0.005

**Table S4:** Summary table from binomial generalized mixed model (logit-link) for analysis
 of parasite prevalence: $PV \sim \text{location} + (1|\text{river})$; PV is the proportion of *Tb*-positive
 samples and the total sample size for each river was used as weights in the model. Model
 run on all data.

Random effects

Groups		Variance	SD
River	(Intercept)	79.49	8.92

N (obs.): 26

N (River): 13

Fixed effects

	Estimate	SE	z	p
(Intercept)	2.01	2.99	0.67	0.502
Location (upstream)	-4.96	0.76	-6.50	< 0.001

**Table S5:** Summary table from binomial generalized mixed model (logit-link) for analysis
 of parasite prevalence: $PV \sim \text{location} + (1|\text{river})$; PV is the proportion of *Tb*-positive
 samples and the total sample size for each river was used as weights in the model. Model
 run on a subset of data, only including *Tb* present rivers.

Random effects

Groups		Variance	SD
River	(Intercept)	32.67	5.72

N (obs.): 18

N (River): 9

Fixed effects

	Estimate	SE	z	p
(Intercept)	6.55	2.65	2.47	0.013
Location (upstream)	-4.97	0.77	-6.49	< 0.001

17 **Table S6:** Summary table from linear mixed model for analysis of parasite load: sqrt(PL)
 18 ~ location + (1|river). Model run on all data.

Random effects

Groups		Variance	SD
River	(Intercept)	5190	72.04
Residual		2749	52.43

N (obs.): 444

N (River): 13

Fixed effects

	Estimate	SE	df	t	p
(Intercept)	82.32	20.31	12.41	4.05	0.002
Location (upstream)	-47.15	5.07	430.60	-9.30	<2e-16

19

20 **Table S7:** Summary table from linear mixed model for analysis of parasite load: $pl \sim$
 21 location + (1|river). Model run on a subset of data, only including *Tb* present rivers.

Random effects

Groups		Variance	SD
River	(Intercept)	5731	75.71
Residual		3571	59.76

N (obs.): 315

N (River): 9

Fixed effects

	Estimate	SE	df	t	p
(Intercept)	115.03	25.68	8.25	4.48	0.002
Location (upstream)	-65.44	6,82	305.26	-9.60	<2e-16

22

23 **Table S8:** Summary table from linear mixed model for analysis of renal hyperplasia
 24 (kidney height, K relative to dorsal musculature height, B): $k \sim b + \text{location} + (1|\text{river})$.
 Model is run on all data without separation of infected and *Tb* absent rivers.

Random effects

Groups		Variance	SD
River	(Intercept)	0.21	0.46
Residual		0.19	0.44

N (obs.): 443

N (River): 13

Fixed effects

	Estimate	SE	df	t	P
(Intercept)	1.12	0.17	38.8	6.4	1.63e-7
Body height	0.04	0.02	440.0	2.2	0.028
Location (upstream)	-0.50	0.04	429.0	-11.8	<2e-16

**Table S9:** Summary table from linear mixed model for analysis of renal hyperplasia
 (kidney height, K relative to dorsal musculature height, B): $k \sim b + \text{location} \times \text{par.pres} +$
 $(1|\text{river})$. Model includes factor parasite presence (par.pres) in interaction with location to
 compare infected and *Tb* present/absent rivers.

Random effects

Groups		Variance	SD
River	(Intercept)	0.15	0.39
Residual		0.18	0.43

N (obs.): 443

N (River): 13

Fixed effects

	Estimate	SE	df	t	p
(Intercept)	0.38	0.24	22.5	1.59	0.125
Body height	0.06	0.02	435.1	3.65	0.0003
Location (upstream)	-0.14	0.08	430.5	-1.70	0.090
Parasite presence (infected river)	0.79	0.24	11.9	3.21	0.008
Location (upstream) × Par. pres. (infected river)	-0.51	0.09	431.1	-5.34	1.51e-7

**Table S10:** Estimated qPCR parameters associated with *Tb* quantification. *Calculations
 were performed with first four ten-fold serial dilution concentrations.

Parameter	
Mean SD for technical replicate Cq values	0.110
Median SD for technical replicate Cq values	0.053
25 percentile for technical replicate Cq values	0.031
75 percentile for technical replicate Cq values	0.113
Plate 1 amplification efficiency %	91.45*
Plate 2 amplification efficiency %	92.16
Plate 3 amplification efficiency %	92.25*
Plate 4 amplification efficiency %	93.20
Plate 5 amplification efficiency %	91.68*
Plate 6 amplification efficiency %	93.60
Plate 1 r^2 of calibration curve	0.999
Plate 2 r^2 of calibration curve	0.993
Plate 3 r^2 of calibration curve	0.999
Plate 4 r^2 of calibration curve	0.997
Plate 5 r^2 of calibration curve	0.999
Plate 6 r^2 of calibration curve	0.998
Synthetic dilution series plate efficiency %	91.78
Synthetic dilution series r^2	0.998